# Memory Efficient Meta-Learning with Large Images

**John Bronskill**[*]
University of Cambridge
jfb54@cam.ac.uk

**Daniela Massiceti**[*]
Microsoft Research
dmassiceti@microsoft.com

**Massimiliano Patacchiola**[*]
University of Cambridge
mp2008@cam.ac.uk

**Katja Hofmann**
Microsoft Research
kahofman@microsoft.com

**Sebastian Nowozin**
Microsoft Research
senowoz@microsoft.com

**Richard E. Turner**
University of Cambridge
ret26@cam.ac.uk

## Abstract

Meta learning approaches to few-shot classification are computationally efficient at test time, requiring just a few optimization steps or single forward pass to learn a new task, but they remain highly memory-intensive to train. This limitation arises because a task's entire support set, which can contain up to 1000 images, must be processed before an optimization step can be taken. Harnessing the performance gains offered by large images thus requires either parallelizing the meta-learner across multiple GPUs, which may not be available, or trade-offs between task and image size when memory constraints apply. We improve on both options by proposing LITE, a general and memory efficient episodic training scheme that enables meta-training on large tasks composed of large images on a single GPU. We achieve this by observing that the gradients for a task can be decomposed into a sum of gradients over the task's training images. This enables us to perform a forward pass on a task's entire training set but realize significant memory savings by back-propagating only a random subset of these images which we show is an unbiased approximation of the full gradient. We use LITE to train meta-learners and demonstrate new state-of-the-art accuracy on the real-world ORBIT benchmark and 3 of the 4 parts of the challenging VTAB+MD benchmark relative to leading meta-learners. LITE also enables meta-learners to be competitive with transfer learning approaches but at a fraction of the test time computational cost, thus serving as a counterpoint to the recent narrative that transfer learning is all you need for few-shot classification.

## 1 Introduction

Meta-learning approaches to few-shot classification are very computationally efficient. Once meta-trained, they can learn a new task at test time with as few as 1-5 optimization steps [1, 2] or a single forward pass through the model [3–5] and with minimal or no hyper-parameter tuning. In contrast, transfer learning approaches based on fine-tuning typically rely on a large pre-trained feature extractor, and instead take 100s-1000s of optimization steps at test time in order to learn a task [6], thus incurring a high computational cost for each new task encountered. This makes meta-learned solutions attractive in compute-constrained deployments, or scenarios where the model must learn multiple different tasks or update on-the-fly (e.g. in continual and online learning settings [7–10]).

However, a crucial barrier to progress is that meta-learning approaches are memory-intensive to train and thus cannot easily leverage large images for a performance boost, as recent fine-tuning approaches have done. This limitation arises because a meta-learner must back-propagate through *all*

---

[*]Authors contributed equally

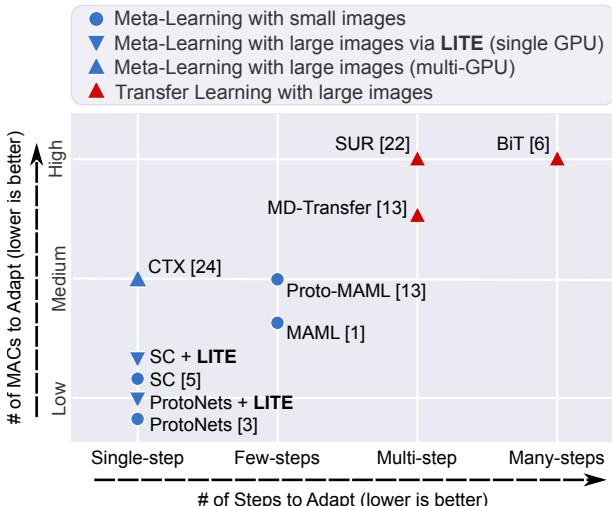

Figure 1: **LITE enables meta-learners to be trained on large images with one GPU thereby significantly improving performance while retaining their test time computational efficiency.** The schematic shows test time efficiency (the number of steps and number of Multiply-Accumulate operations (MACs) needed to learn a new task at test time) and whether the method can be trained on large images (required for good performance). Existing meta-learners are cheap to adapt but trained on small images (multiple GPUs are required for large images), transfer learning methods are expensive to adapt but trainable on large images. Meta-learners with LITE get the best of both worlds. Note: SC + LITE is Simple CNAPs [5] trained with LITE.

the examples in a task's support set (i.e. the task's training set) that contribute to a prediction on a query example. In some cases, this can be as many as 1000 images [11]. As a result, the amount of memory required for the computational graph grows linearly with the number of support images, and quadratically with their dimension. In contrast, transfer learning approaches can employ standard batch processing techniques to scale to larger images when under memory constraints — a feature which has contributed significantly to their recent success on few-shot benchmarks [11].

Current solutions for training meta-learners on large images include 1) parallelizing the model across multiple GPUs, which may not be available or convenient, 2) considering tasks with fewer support images, and 3) employing gradient/activation checkpointing methods [12] which incur longer training times and still fall short of the task sizes required in key benchmarks. Instead, most existing work [1, 3, 2, 13, 4, 14] has opted for training on large tasks but small images which translates poorly into real-world applications and limits competitiveness on few-shot benchmarks.

In this work, we improve on these alternatives by proposing LITE, a Large Image *and* Task Episodic training scheme for meta-learning models that enables training with large images and large tasks, on a single GPU. We achieve this through the simple observation that meta-learners typically aggregate a task's support examples using a permutation invariant sum operation. This structure ensures invariance to the ordering of the support set. Consequently, the gradients for a task can be decomposed as a sum of gradient contributions from the task's support examples. This enables us to perform a forward pass on a task's entire support set, but realize significant savings in memory by back-propagating only a random subset of these images which we show is an unbiased approximation of the true gradient. Fig. 1 illustrates the key trade-offs, with LITE enabling meta-learners to benefit from large images for improved classification accuracy but still remain computationally efficient at test time.

We use LITE to train key meta-learning methods and show that our best performing instantiation – Simple CNAPs [5] with LITE – achieves state-of-the-art results relative to all meta-learners on two challenging few-shot benchmarks: VTAB+MD [11], an extensive suite of both meta-learning and transfer learning tasks, and ORBIT [14], an object recognition benchmark of high-variation real-world videos. Our results showcase the unique advantage of meta-learning methods – that when properly trained they can be competitive with transfer learning approaches in terms of accuracy for a fraction of the computational cost at test time — and they serve as a counterpoint to the recent narrative that transfer learning is all you need for few-shot classification.

**Our contributions**

1. LITE, a general and memory-efficient episodic training scheme which enables meta-learning models to be trained on large images and large tasks on a single GPU.
2. A mathematical justification for approximating the true gradient with a random subset of a task's support examples which applies to common classes of meta-learning methods.
3. Instantiations of LITE on key classes of meta-learners to demonstrate its versatility.

4. State-of-the-art performance using Simple CNAPs with LITE compared to other leading meta-learners on two challenging few-shot benchmarks, VTAB+MD and ORBIT [2]

## 2 Why Meta-Learning with Large Images and Tasks is Difficult

**Meta-learning preliminaries** In few-shot image classification, the goal is to recognize new classes when given only a few training (or support) images of each class. Meta-learners typically achieve this through *episodic* training [15]. Here, an episode or task $\tau$ contains a support set $\mathcal{D}_S^\tau = \{(\boldsymbol{x}_n^\tau, y_n^\tau)\}_{n=1}^{N_\tau}$ and a query set $\mathcal{D}_Q^\tau = \{(\boldsymbol{x}_m^{\tau*}, y_m^{\tau*})\}_{m=1}^{M_\tau}$, where $(\boldsymbol{x}, y)$ is an image-label pair, $N_\tau$ is the number of (labeled) support elements given to learn the new classes, and $M_\tau$ is the number of query elements requiring predictions. Note that in a given task, elements in $\mathcal{D}_Q^\tau$ are drawn from the same set of classes as the elements in $\mathcal{D}_S^\tau$. For brevity we may use the shorthand $\mathcal{D}_S = \{\boldsymbol{x}, y\}$ and $\mathcal{D}_Q = \{\boldsymbol{x}^*, y^*\}$.

During meta-training, a meta-learner is exposed to a large number of training tasks $\{\tau\}$. For each task $\tau$, the meta-learner takes as input the support set $\mathcal{D}_S$ and outputs the parameters of a classifier that has been adapted to the current task $\boldsymbol{\theta}^\tau = \boldsymbol{\theta}_\phi(\mathcal{D}_S)$. The classifier can now make task-specific probabilistic predictions $f(\boldsymbol{x}^*, \boldsymbol{\theta}^\tau = \boldsymbol{\theta}_\phi(\mathcal{D}_S))$ for any query input $\boldsymbol{x}^*$ (see Fig. 2). A function $\mathcal{L}(y^*, f(\boldsymbol{x}^*, \boldsymbol{\theta}^\tau))$ computes the loss between the adapted classifier's predictions for the query input and the true label $y^*$ which is observed during meta-training. Assuming that $\mathcal{L}$, $f$, and $\boldsymbol{\theta}_\phi(\mathcal{D}_S)$ are differentiable, the meta-learner can then be trained with stochastic gradient descent by back-propagating the loss and updating the parameters $\phi$.

At meta-test time, the trained meta-learner is given a set of unseen test tasks, which typically contain classes that have *not* been seen during meta-training. For each task, the meta-learner is given its support set $\mathcal{D}_S$, and is then evaluated on its predictions for all the query inputs $\boldsymbol{x}^*$ (Fig. 2, left).

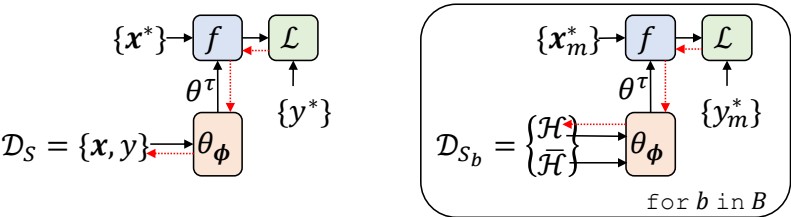

Figure 2: Left: Canonical meta-learner. Right: Meta-learner with LITE. The red dotted line shows the back-propagated gradients. Refer to Algorithm 1 for nomenclature.

**Large memory requirements for meta-training** The primary bottleneck to using large images (i.e. $\geq 224 \times 224$ pixels) in meta-learning approaches is the large amount of (GPU) memory required to process a task's support set $\mathcal{D}_S$ during meta-training. Specifically, the meta-learner $\boldsymbol{\theta}_\phi(\mathcal{D}_S)$ must perform a forward pass with a task's *entire* support set before it can back-propagate the loss for query elements $(\boldsymbol{x}^*, y) \in D_Q$ (and release the computation graph), thus preventing the use of conventional batch processing. The amount of memory required scales linearly with the number of support images $N^\tau$ and quadratically with their dimensions. If $N^\tau$ is large (e.g. the recent VTAB+MD benchmark [11] requires a task's support set to be as large as 1000 images), memory on a single GPU is thus quickly exceeded for large images.

Note, the number of query elements in the task $M^\tau$ is not a bottleneck when using large images as the loss decomposes over elements of the query set $\mathcal{D}_Q$ and is therefore amenable to mini-batching. By contrast, as the classifier itself is a non-linear function of the *support set*, the loss does not decompose and so it is not obvious how to apply similar ideas to allow scaling of $\mathcal{D}_S$ in a principled way.

Current ad hoc solutions to this problem are: (i) parallelize the meta-learner across multiple GPUs which may not be convenient or available and can involve significant engineering effort; (ii) train on tasks with smaller (or sub-sampled) support sets which may adversely affect performance on test tasks with more classes and/or large numbers of samples per class; (iii) train on tasks with smaller images (e.g. $84 \times 84$ pixels in *mini*ImageNet [16]) which limits performance and translates poorly to many real-world applications; or (iv) trade memory usage for additional computation [12] by

---

[2]Source code for ORBIT experiments is available at `https://github.com/microsoft/ORBIT-Dataset` and for the VTAB+MD experiments at `https://github.com/cambridge-mlg/LITE`.

**Algorithm 1** LITE for a meta-training task $\tau$

---

**Require:** $\mathcal{D}_S$: task support set; $\mathcal{D}_Q$: task query set; $N$: number of support examples in $\mathcal{D}_S$; $H$: number of elements in $\mathcal{D}_S$ to back-propagate; $M$: number of query examples in $\mathcal{D}_Q$; $M_b$: batch size for $\mathcal{D}_Q$; $\texttt{backward()} \equiv$ function to back-propagate a loss; $\texttt{step()} \equiv$ function to update parameters with a gradient step.

1: $B \leftarrow \text{ceil}(M/M_b)$          ▷ number of query batches
2: **for all** $b \in 1, \ldots, B$ **do**
3:      $\mathcal{D}_{Q_b} \leftarrow \{\boldsymbol{x}_m^*, y_m^*\}_{m=1}^{M_b}$          ▷ get query batch from $\mathcal{D}_Q$
4:      $\mathcal{H} \leftarrow \{(\boldsymbol{x}_{n_h}, y_{n_h})\}_{h=1}^{H}$ where $\{n_h\}_{h=1}^{H} \sim \mathcal{U}(1, N)$          ▷ $\mathcal{H}$ to back-propagate
5:      $\overline{\mathcal{H}} \leftarrow \mathcal{D}_S \cap \mathcal{H}$          ▷ $\overline{\mathcal{H}}$ to not back-propagate
6:      $\mathcal{D}_{S_b} \leftarrow \mathcal{H} \cup \overline{\mathcal{H}}$
7:      $\boldsymbol{\theta}^\tau \leftarrow \boldsymbol{\theta}(\mathcal{D}_{S_b})$
8:      $L_b \leftarrow \frac{1}{M_b} \Sigma_{m=1}^{M_b} \mathcal{L}(y_m^*, f(\boldsymbol{x}_m^*, \boldsymbol{\theta}^\tau))$          ▷ get loss of query batch
9:      $\texttt{backward}(L_b)$          ▷ back-propagate loss on query batch
10: **end for**
11: $\boldsymbol{\phi} \leftarrow \texttt{step}(\boldsymbol{\phi}, N/H)$          ▷ update $\boldsymbol{\phi}$ using weighting factor $N/H$

---

employing *activation/gradient checkpointing* (i.e. during training store only a subset of intermediate activations in a network needed for backpropagation and recompute the rest with additional forward computations when needed) which allows for training on larger tasks at the expense of training time, but still falls well short of the memory needed to accommodate the task sizes required for key benchmarks (e.g. VTAB+MD).

Although training meta-learners has large memory requirements, meta-testing is generally memory efficient, requiring only a small number of gradient operations, or none at all, compared to transfer learning approaches that would perform large numbers of gradient-based updates at test time.

## 3 Large Image and Task Episodic (LITE) training

In this section, we introduce our general and memory-efficient solution for training meta-learners episodically on tasks with large support sets and large images. We call our approach *Large Image and Task Episodic* training or LITE. In Section 3.1, we describe how LITE can be applied to key classes of meta-learners.

**Approach** The fundamental idea underlying LITE is to perform a forward pass using the entire support set $\mathcal{D}_S$, but to compute the gradient contribution on only a small random subset of the examples in the support set. By doing this, we realize large savings in memory that includes gradients, activations, and the computation graph for the elements of $\mathcal{D}_S$ that are not back-propagated. This is an approximation of the true gradient that would result if back-propagation was performed on all of the examples in $\mathcal{D}_S$. In the following section, we show that this approximation is an unbiased estimate of the true gradient. The approach for a general meta-learner is detailed in Algorithm 1 and shown diagrammatically in Fig. 2.

**Mathematical justification** The parameters of the meta-learner $\phi$ are found by minimizing the expected loss over all tasks.

$$\underset{\boldsymbol{\phi}}{\text{argmin}} \sum_{\tau=1}^{T} \sum_{m=1}^{M_\tau} \mathcal{L}\left(y_m^{\tau*}, f\left(\boldsymbol{x}_m^{\tau*}, \boldsymbol{\theta}_{\boldsymbol{\phi}}(\mathcal{D}_S^\tau)\right)\right). \tag{1}$$

In most meta-learning approaches, the support set enters into the loss through a sum over the $N$ individual contributions from each data point it contains. This structure enables the meta-learners to be invariant to the ordering of the support set and allows all members of the support set to contribute to the adapted parameters (unlike alternative permutation invariant operators like *max* or *min*). Below we show in blue how this sum arises in popular brands of meta-learners.

In amortization methods (e.g. CNAPs [4] and VERSA [17]), the aggregation of support set points is built in directly via a deep set encoder $e_{\phi_1}(\cdot)$. This encodes the support set into an embedding vector

which is mapped to the classifier parameters by a hyper-network $t_{\phi_0}(\cdot)$.

$$\boldsymbol{\theta}_{\boldsymbol{\phi}}(\mathcal{D}_S) = t_{\phi_0}\left(\sum_{n=1}^{N} e_{\phi_1}(\boldsymbol{x}_n, y_n)\right). \tag{2}$$

In gradient-based methods (e.g. MAML [1]), the classifier parameters are adapted from an initial value $\phi_0$ using a sum of derivatives of an inner-loop loss computed for each data point in the support set. The derivatives play the role of the deep set encoder in amortization methods.

$$\boldsymbol{\theta}_{\boldsymbol{\phi}}(\mathcal{D}_S) = \phi_0 + \phi_1 \sum_{n=1}^{N} \frac{d}{d\boldsymbol{\phi}}\mathcal{L}_{\text{inner}}(y_n, f(\boldsymbol{x}_n, \boldsymbol{\phi}))\Big|_{\boldsymbol{\phi}=\phi_0} \tag{3}$$

Metric-based methods (e.g. ProtoNets [3]) comprise a body formed of a feature extractor and a head formed from a distance-based classifier. The classifier's body parameters are not adapted in a task specific way $\boldsymbol{\theta}_{\boldsymbol{\phi},c}^{(\text{body})}(\mathcal{D}_S) = \phi_0$. The classifier's head is adapted by averaging the activations for each class in the support set to form prototypes. Letting $k_c$ denote the number of support examples of class $c$, the adapted head parameters are given by

$$\boldsymbol{\theta}_{\boldsymbol{\phi},c}^{(\text{head})}(\mathcal{D}_S) = \frac{1}{k_c}\sum_{i=1}^{k_c} f(\boldsymbol{x}_i^{(c)}, \phi_0) = \frac{1}{k_c}\sum_{n=1}^{N} \mathbb{1}(y_n = c)f(\boldsymbol{x}_n, \phi_0). \tag{4}$$

Query points can then be classified using their distance from these prototypes $d(f(\boldsymbol{x}_n^*, \phi_0), \boldsymbol{\theta}_{\boldsymbol{\phi},c}^{(\text{head})})$.

We have established that in many meta-learners, each support set affects the classifier parameters and therefore the loss through a sum of contributions from each of its elements. We now focus on the consequences of this structure on the gradients of the loss with respect to the meta-learner's parameters. To reduce clutter, we consider the contribution from just a single query point from a single task and suppress the dependence of the loss on the classifier and the query data point, writing

$$\mathcal{L}\left(y^*, f(\boldsymbol{x}^*, \boldsymbol{\theta}_{\boldsymbol{\phi}}(\mathcal{D}_S))\right) = \mathcal{L}\left(e_{\boldsymbol{\phi}}(\mathcal{D}_S)\right) \text{ where } e_{\boldsymbol{\phi}}(\mathcal{D}_S) = \sum_{n=1}^{N} e_{\boldsymbol{\phi}}(\boldsymbol{x}_n, y_n) = \sum_{n=1}^{N} e_{\boldsymbol{\phi}}^{(n)}. \tag{5}$$

As a consequence of the summation, the derivative of the loss is given by

$$\frac{d}{d\boldsymbol{\phi}}\mathcal{L}(e_{\boldsymbol{\phi}}(\mathcal{D}_S)) = \mathcal{L}'(e_{\boldsymbol{\phi}}(\mathcal{D}_S)) \times \left(\sum_{n=1}^{N} \frac{de_{\boldsymbol{\phi}}^{(n)}}{d\boldsymbol{\phi}}\right) \text{ where } \mathcal{L}'(e_{\boldsymbol{\phi}}(\mathcal{D}_S)) = \frac{d\mathcal{L}(e)}{de}\Big|_{e=e_{\boldsymbol{\phi}}(\mathcal{D}_S)} \tag{6}$$

which is a product of the sensitivity of the loss to the encoding of the data points and the sensitivity of the contribution to the encoding from each data point w.r.t. the meta-learner's parameters. This second term is the source of the memory overhead when training meta-learners, but importantly, it can be rewritten as an expectation w.r.t. a uniform distribution over the support set data-point indices,

$$\frac{d}{d\boldsymbol{\phi}}\mathcal{L}(e_{\boldsymbol{\phi}}(\mathcal{D}_S)) = N \, \mathcal{L}'(e_{\boldsymbol{\phi}}(\mathcal{D}_S)) \, \mathbb{E}_{n\sim\mathcal{U}(1,N)}\left[\frac{de_{\boldsymbol{\phi}}^{(n)}}{d\boldsymbol{\phi}}\right]. \tag{7}$$

We can now define the LITE estimator of the loss-derivative by approximating the expectation by Monte Carlo sampling $H$ times,

$$\frac{d}{d\boldsymbol{\phi}}\mathcal{L}(e_{\boldsymbol{\phi}}(\mathcal{D}_S)) \approx \frac{N}{H}\mathcal{L}'(e_{\boldsymbol{\phi}}(\mathcal{D}_S))\sum_{h=1}^{H} \frac{de_{\boldsymbol{\phi}}^{(n_h)}}{d\boldsymbol{\phi}} = \frac{d}{d\boldsymbol{\phi}}\hat{\mathcal{L}}(e_{\boldsymbol{\phi}}(\mathcal{D}_S)) \text{ where } \{n_h\}_{h=1}^{H} \sim \mathcal{U}(1,N). \tag{8}$$

This estimator is unbiased, converging to the true gradient as $H \to \infty$. The estimator does not simply involve subsampling of the support set – parts of it depend on all the support set data points $\mathcal{D}_S$ – and this is essential for it to be unbiased. The expectation and variance of this estimator are

$$\mathbb{E}_{\{n_h\}\sim\mathcal{U}(1,N)}\left[\frac{d\hat{\mathcal{L}}}{d\boldsymbol{\phi}}\right] = \frac{d\mathcal{L}}{d\boldsymbol{\phi}} \text{ and } \mathbb{V}_{\{n_h\}\sim\mathcal{U}(1,N)}\left[\frac{d\hat{\mathcal{L}}}{d\boldsymbol{\phi}}\right] = \frac{N^2}{H}(\mathcal{L}')^2 \, \mathbb{V}_{\{n_h\}\sim\mathcal{U}(1,N)}\left[\frac{de_{\boldsymbol{\phi}}^{(n_h)}}{d\boldsymbol{\phi}}\right].$$

In Section 5.3, we empirically show that the LITE gradient estimate is unbiased and that its standard deviation is smaller than that of the naive estimator formed by sub-sampling the full support set. LITE provides memory savings by subsampling $H$ examples from the support set, with $H < N$, and back-propagating only them. Crucially, a forward pass is still performed with the complementary set of points, with cardinality $N - H$, but these are not back-propagated.

## 3.1 Applying LITE to key meta-learning approaches

To demonstrate its versatility, we now describe how to apply LITE to models within some of the main classes of meta-learners: CNAPs [4] and Simple CNAPs [5] for amortization-based methods and ProtoNets [3] for metric-based methods. Note, these are a few possible instantiations. LITE can be applied to other meta-learning methods in a straightforward manner.

In the descriptions below, we consider just one query batch $\mathcal{D}_{Q_b}$ (i.e. one iteration of the for-loop in Algorithm 1). Note, in practice, whenever $\mathcal{H}$ is passed through a module, back-propagation is enabled, while for $\overline{\mathcal{H}}$, back-propagation is disabled.[3] Furthermore, since typically $|\mathcal{H}| \ll |\overline{\mathcal{H}}|$, we can forward $\mathcal{H}$ in a single batch, however, we need to split $\overline{\mathcal{H}}$ into smaller batches. Since $\overline{\mathcal{H}}$ does not require gradients to be computed, this can be done without a significant impact on memory.

**CNAPs [4], Simple CNAPs [5] + LITE (Appendix A.1)** CNAPs variants are amortization-based methods whose hyper-networks take a task's support set as input and generate FiLM layer [18] parameters which modulate a fixed feature extractor. The classifier head can also be generated (CNAPs [4]) or adopt a metric-based approach (Simple CNAPs [5]), thus both variants can be adapted with just a single forward pass of the support set at test time. Meta-training them with LITE involves passing $\mathcal{H}$ and then $\overline{\mathcal{H}}$ through their set-encoder $e_{\phi_1}$, and then averaging all the low-dimensional embeddings to get an embedding for the task. The task embedding is then input into a set of MLPs which generate FiLM layer parameters. $\mathcal{H}$ is passed through this configured feature extractor, followed by $\overline{\mathcal{H}}$, to get the task-adapted features for all support examples in $\mathcal{D}_{S_b}$. For CNAPs, the task-adapted features of $\mathcal{H}$ and $\overline{\mathcal{H}}$ are pooled by class and fed into a second MLP which generates the parameters of the fully-connected classification layer. For Simple CNAPs, the task-adapted features of $\mathcal{H}$ and $\overline{\mathcal{H}}$ are instead used to compute class-wise distributions (i.e. class mean and covariance matrices). With back-propagation enabled, the query batch $\mathcal{D}_{Q_b}$ is then passed through the task-configured feature extractor and classified with the task-configured classifier (for CNAPs), or with the Mahalanobis distance [19] to the class-wise distributions (for Simple CNAPs). The query batch loss is computed, and only back-propagated for $\mathcal{H}$. Note that the feature extractor is pre-trained and frozen, and only the parameters of the set-encoder and generator MLPs are learned.

**ProtoNets [3] + LITE (Appendix A.2)** ProtoNets [3] is a metric-based approach which computes a set of class prototypes from the support set and then classifies query examples by their (e.g. Euclidean) distance to these prototypes. Like CNAPs variants, it requires only a single forward pass to learn a new task. Meta-training ProtoNets with LITE involves passing $\mathcal{H}$ through the feature extractor with back-propagation enabled, followed by $\overline{\mathcal{H}}$ with back-propagation disabled, to obtain features for all support examples $\mathcal{D}_{S_b}$. These features are averaged by class to compute the prototypes such that (with back-propagation enabled) the query batch $\mathcal{D}_{S_b}$ can be passed through the feature extractor and classified based on the Euclidean distance. The loss of the query batch is computed and only back-propagated for $\mathcal{H}$. Note that here all the parameters of the feature extractor are learned.

## 4 Related work

We review the two main approaches to few-shot learning: transfer learning methods which are easy to scale to large images but are costly to adapt at test time, and meta-learning methods which are harder to scale but cheap to adapt (see Fig. 1). Note, we do not cover methods already described above.

Transfer learning approaches have demonstrated state-of-the-art performance on challenging few-shot benchmarks [20, 11]. However, they incur a high computational cost at test time as they rely on large (pre-trained) feature extractors which are fine-tuned with many optimization steps. MD-Transfer [13] fine-tunes all the parameters in a ResNet18 feature extractor with a cosine classifier head for 200 optimization steps. BiT [6] fine-tunes a feature extractor (pre-trained on 300M images in the JFT-300M dataset [21]) with a linear head, in some cases for up to 20,000 optimization steps to achieve its state-of-the-art results on the VTAB [20] benchmark. SUR [22] instead trains 7 ResNet-50 feature extractors, one for each training dataset. At test time, the predictions from each are concatenated and fine-tuned to minimize a cosine-similarity loss. All of these approaches involve on the order of teras to petas of Mulitply-Accumulate operations (MACs) to learn a single new test task, and this must

---

[3]In PyTorch, this can be achieved by setting `torch.grad.enabled = True` when passing $\mathcal{H}$, and `torch.grad.enabled = False` when passing $\overline{\mathcal{H}}$.

be repeated for each new task encountered. Furthermore, for each new task type, transfer learning approaches may need to be tuned on a validation set to obtain the optimal hyper-parameters.

In comparison, meta-learning approaches [23] generally require orders of magnitude fewer MACs and steps to learn a new task at test time. Popular approaches include CNAPs [4], Simple CNAPs [5], ProtoNets [3], and MAML [1] and are discussed in Section 3.1. Others include ProtoMAML [13] which fuses ProtoNets and MAML by initializing the classifier weights with the prototypes and then, like MAML, takes a few optimization steps to tune the weights to the task. It therefore incurs a similar cost to adapt as MAML, except it must additionally compute the prototypes. Finally, CTX [24] replaces the final average pooling layer of ProtoNets with a transformer layer that generates a series of prototypes which are spatially aware and aligned with the task. Like CNAPs, it requires just a single forward pass, however, requires more MACs to adapt since it uses a larger feature extractor (ResNet-34) and an attention module. CTX is one of the few meta-learning approaches that has been meta-trained on $224 \times 224$ images, but requires 7 days of training on 8 GPUs.

Memory efficient variants of MAML have been developed. First-order MAML [1] saves memory by avoiding the estimate of second-order derivatives. This is also done in Reptile [25] that additionally avoids unrolling the computation graph, performing standard gradient descent at each adaptation step. Implicit MAML [26], decouples the meta-gradient from the inner loop and is able to handle many gradient steps without memory constraints. In addition [27], proposes methods to reduce the computation overhead of meta-training MAML for large tasks. Note that, in all these cases, savings arise from working around the limitations of MAML, while LITE is more general.

## 5 Experiments

In this section, we demonstrate that meta-learners trained with LITE achieve state-of-the-art performance among meta-learners on two challenging few-shot classification benchmarks: (i) ORBIT [14] which is a real-world few-shot object recognition dataset for teachable object recognizers; and (ii) VTAB+MD [11] which is composed of the Visual Task Adaptation Benchmark (VTAB) [20] and Meta-Dataset (MD) [13] and combines both few-shot and transfer learning tasks. We compare LITE meta-learners with state-of-the-art meta-learning and transfer learning methods in terms of classification accuracy, computational cost/time to learn a new task, and number of model parameters.

### 5.1 ORBIT Teachable Object Recognition Benchmark

Table 1: **Training meta-learners on large images with LITE achieves state-of-the-art accuracy with low test time adaption cost on ORBIT.** Results are reported as the average (95% confidence interval) over 85 test tasks (5 tasks per test user, 17 test users). $I$ is image size. $f$ is model trained with/without LITE. RN-18 is ResNet-18. EN-B0 is EfficientNet-B0. T is $\times 10^{12}$ MACs. F is forward pass. FB is forward-backward pass. Time is average wall clock time per task in seconds.

| MODEL | $I$  $f$ | Clean Videos | | Clutter Videos | | Test-time adaption | | | |
|---|---|---|---|---|---|---|---|---|---|
| | | FRAME ACC ↑ | VIDEO ACC ↑ | FRAME ACC ↑ | VIDEO ACC ↑ | MACS ↓ | STEPS ↓ | TIME ↓ | PARAMS ↓ |
| FineTuner [28] | 84 RN-18 | 69.5 (2.2) | 79.7 (2.6) | 53.7 (1.8) | 63.1 (2.4) | 317.70T | 50FB | 53.94s | 11.17M |
| | 224 RN-18 | 72.2 (2.2) | 81.9 (2.6) | 56.7 (2.0) | 61.3 (2.5) | 546.57T | 50FB | 96.23s | 11.18M |
| | 224 EN-B0 | **78.1 (2.0)** | 85.9 (2.3) | **63.1 (1.8)** | 66.9 (2.4) | 121.02T | 50FB | 139.99s | **4.01M** |
| MAML [1] | 84 RN-18 | 70.6 (2.1) | 80.9 (2.6) | 51.7 (1.9) | 57.9 (2.5) | 95.31T | 15FB | 36.98s | 11.17M |
| | 224 RN-18 | 75.7 (1.9) | 86.1 (2.3) | 59.3 (1.9) | 64.3 (2.4) | 163.97T | 15FB | 65.22s | 11.18M |
| | 224 EN-B0 | **79.3 (1.9)** | 87.5 (2.2) | **64.6 (1.9)** | 69.4 (2.3) | 36.31T | 15FB | 117.89s | **4.01M** |
| ProtoNets [3] | 84 RN-18 | 65.2 (2.0) | 81.9 (2.5) | 50.3 (1.7) | 59.9 (2.5) | 3.18T | **1F** | **0.73s** | 11.17M |
| | 224 RN-18 + LITE | 76.7 (1.9) | 86.4 (2.2) | 61.4 (1.8) | **68.5 (2.4)** | 5.47T | **1F** | 1.07s | 11.18M |
| | 224 EN-B0 + LITE | 82.1 (1.7) | 91.2 (1.9) | 66.3 (1.8) | 72.9 (2.3) | 1.21T | **1F** | 1.72s | **4.01M** |
| CNAPs [4] | 84 RN-18 | 66.2 (2.1) | 79.6 (2.6) | 51.5 (1.8) | 59.5 (2.5) | 3.48T | **1F** | 0.98s | 12.75M |
| | 224 RN-18 + LITE | 76.0 (1.9) | 84.9 (2.3) | 58.2 (1.9) | 62.5 (2.5) | 7.64T | **1F** | 2.11s | 12.76M |
| | 224 EN-B0 + LITE | 79.6 (1.9) | 87.6 (2.2) | 63.3 (1.9) | 69.2 (2.3) | 3.38T | **1F** | 2.85s | 10.59M |
| Simple CNAPs [5] | 84 RN-18 | 70.3 (2.1) | 83.0 (2.5) | 53.9 (1.8) | 62.0 (2.5) | 3.48T | **1F** | 1.01s | 11.97M |
| | 224 RN-18 + LITE | 76.5 (2.0) | 86.4 (2.2) | 57.5 (1.9) | 64.6 (2.4) | 7.64T | **1F** | 2.14s | 11.97M |
| | 224 EN-B0 + LITE | **82.7 (1.7)** | **91.8 (1.8)** | **65.6 (1.9)** | **71.9 (2.3)** | 3.39T | **1F** | 2.92s | 5.67M |

ORBIT [14] is a highly realistic few-shot video dataset collected by people who are blind/low-vision. It presents an object recognition benchmark task which involves personalizing (i.e. adapting) a

recognizer to each individual user with just a few (support) videos they have recorded of their objects. To achieve this, the benchmark splits data collectors into disjoint train, validation, and test user sets along with their corresponding objects and videos. Models are then meta-trained on the train users, and meta-tested on how well they can learn a test user's objects given just their videos (on a user-by-user basis). The benchmark has two evaluation modes: how well the meta-trained model can recognize a test user's objects in *clean* videos where there is only that object present, and in *clutter* videos where that object appears within a realistic, multi-object scene.

**Experiments**   We meta-train ProtoNets [3], CNAPs [4] and Simple CNAPs [5] with LITE on tasks composed of large ($224 \times 224$) images. We also meta-train first-order MAML on large images as a baseline. Since first-order MAML can process task support sets in batches, we simply reduce the batch size and do not need to use LITE. We compare all of the above to meta-training on tasks of small ($84 \times 84$) images (i.e. the original baselines [14]). We also include a transfer learning approach, FineTuner [28], which freezes a pre-trained feature extractor and fine-tunes just the linear classifier for 50 optimization steps. For each model, we consider a ResNet-18 (RN-18) and EfficientNet-B0 (EN-B0) feature extractor, both pre-trained on ImageNet [29]. We follow the task sampling protocols described in [14] (see Appendices B and C.1 for details). We also include analyses on meta-training with small tasks of large images in Appendix D.3.

**Results**   In Table 1, we report frame accuracy and video accuracy, averaged over all the query videos from all tasks across all test users (17 test users, 85 tasks in total), along with their corresponding 95% confidence intervals. We also report the computational cost to learn a new task at test time in terms of the number of Multiply-Accumulate operations (MACs), the number of steps to adapt, and the wall clock time to adapt in seconds. See Appendix C.1 for results on additional metrics. The key observations from our results are:

- Training on larger ($224 \times 224$) images leads to better performance compared to smaller ($84 \times 84$) images. The boost is significant for both clean and clutter videos, though absolute performance remains lower on clutter videos. This suggests that object detection or other attention-based mechanisms may be required to further exploit large images in more complex/cluttered scenes.
- All meta-learners + LITE set a new state-of-the-art on clean videos, and perform competitively with the FineTuner on cluttered videos, using an EfficientNet-B0 backbone.
- Meta-learners are competitive with transfer learning approaches in accuracy but are almost two orders of magnitude more efficient in the number of MACs and the time to learn a new task, and one order of magnitude smaller in the number of steps to adapt.

## 5.2   VTAB+MD

VTAB+MD [11] combines revised versions of the Meta-Dataset (MD) and VTAB datasets and is one of the largest, most comprehensive, and most challenging benchmarks for few-shot learning systems. The MD-v2 part of the benchmark involves testing on 8 diverse datasets while VTAB-v2 involves testing on 18 datasets grouped into three different cataegories (natural, specialized, and structured).

**Results**   Fig. 3 compares our best meta-learner, Simple CNAPs + LITE, with 6 other competitive approaches on VTAB+MD: BiT, MD-Transfer, and SUR are transfer learning based methods while ProtoMAML, ProtoNets, and CTX are meta-learning based. Note, comparisons are not always like-for-like as methods use differing backbones, image sizes, and pre-training datasets. Appendix D.2 provides this information along with the precise numbers and 95% confidence intervals. Appendices B and C.2 summarize all implementation and experimental details, and Appendix D.3 includes further analyses on the impact of task size on meta-training. The key observations from our results are:

- On MD-v2, Simple CNAPs + LITE has the highest average score and sets a new state-of-the-art.
- On VTAB-v2, BiT scores highest overall, but among meta-learners, Simple CNAPs + LITE is the best overall and on the natural and specialized sections. It falls short of CTX on the structured section due to poor performance on dSprites which involves predicting the position and orientation of small white shapes on a black background – a task quite different to image classification.
- These results are significant given that CTX takes 7 days to train on 8 GPUs, whereas Simple CNAPs + LITE trains in about 20 hours on a single 16GB GPU. In addition, Simple CNAPs + LITE uses a relatively small pre-trained backbone (4.0M parameters) compared to SUR's 7 pre-trained ResNet-50s (one for each MD-v2 training set, plus ImageNet).

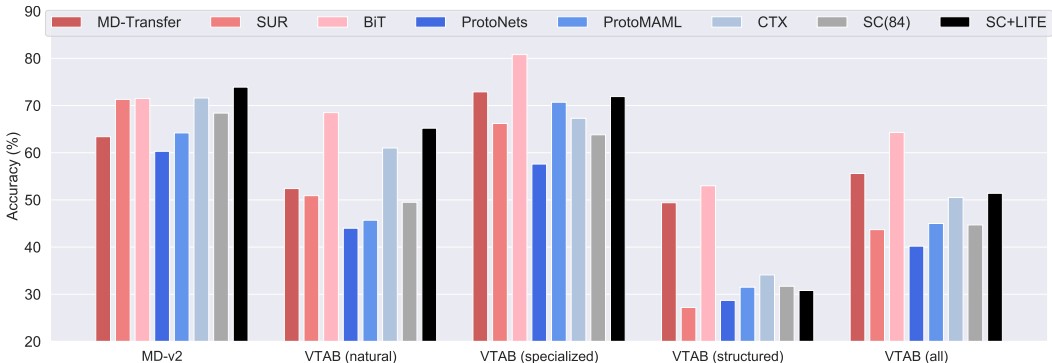

Figure 3: Summary of results on VTAB+MD. **Simple CNAPs with LITE (SC + LITE, black bar) trained on** $224 \times 224$ **images achieves state-of-the-art accuracy on Meta-Dataset (MD-v2), and state-of-the-art accuracy among meta-learners on 3 of 4 parts of VTAB**. As reference, we have included transfer learning methods (red bars), other meta-learning methods (blue bars), and Simple CNAPs without LITE trained on small images ($84 \times 84$, gray bar). Competitive results from [11]. See Table D.2 for tabular results on individual datasets.

- Simple CNAPs + LITE using 224×224 images significantly outperforms Simple CNAPs using 84×84 images, except for when the dataset images are small (e.g. Omniglot, QuickDraw, dSprites). This demonstrates that using large images and an approximation to the support set gradients achieves superior results compared to using small images with exact gradients.
- MD-Transfer and BiT perform strongly across VTAB+MD, however, Simple CNAPs + LITE is significantly faster to adapt to a new task requiring only a forward pass of the support set with no hyper-parameter tuning whatsoever. The transfer learners instead perform 100s of optimization steps to adapt to a new task and may require a human-in the-loop to tune hyper-parameters such as the learning rate and number of optimization steps.

### 5.3  Effect of varying $|\mathcal{H}|$

Table 2 shows the effect of varying $|\mathcal{H}|$, the number of examples back-propagated per task, on the VTAB+MD benchmark. Performance is consistent across different $|\mathcal{H}|$, an expected result since LITE provides an unbiased estimate of the true gradient. For both Simple CNAPs and ProtoNets + LITE, the results at the lowest values of $|\mathcal{H}|$ are respectable, though they fall short of what can be achieved at $|\mathcal{H}| = 40$. Thus, the support gradient information does improve the solution, albeit by only 1-2 percentage points. Note, we report the lowest setting as $|\mathcal{H}| = 1$ for Simple CNAPs but $|\mathcal{H}| = 0$ for ProtoNets. This is because Simple CNAPs's adaptation network (which processes the support set) shares no parameters with the feature extractor and thus will not be learned if the support gradients are completely ignored. On the other hand, ProtoNets' adaptation network shares all its parameters with the feature extractor, thus can be meta-learned even when the support gradients are neglected.

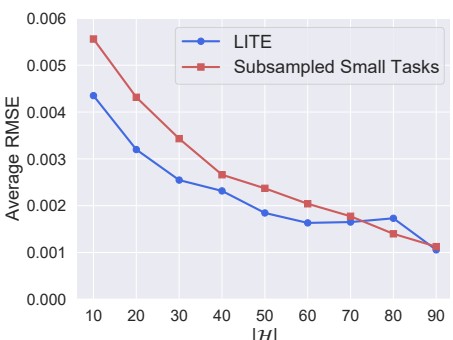

Figure 4: Average Root Mean Square Error (RMSE) w.r.t. the true gradients versus $|\mathcal{H}|$ for LITE and sub-sampled tasks on 84×84 images (10-way, 10-shot, $|\mathcal{D}_S| = 100$).

Finally, in the two rightmost columns, we compare the classification accuracy for $|\mathcal{H}| = |\mathcal{D}_S|$ (i.e. using the full support set gradient) to $|\mathcal{H}| = 40$ (i.e. using LITE). Due to memory constraints, we do this at image size $84 \times 84$. Here we see that the difference in accuracy is significant. We expect that performance will smoothly interpolate as $|\mathcal{H}|$ is increased from 40 to the size of the largest support set (at which point the full gradient is computed). This validates how LITE can be used to trade-off GPU memory usage for classification accuracy by varying $|\mathcal{H}|$. Furthermore, we conduct an empirical analysis (see Table D.7) which shows that the LITE gradient estimates and the gradients when using smaller sub-sampled tasks are unbiased w.r.t. the true gradients. However, Fig. 4 shows that LITE offers a significantly lower root mean square error w.r.t. the true gradients compared to using sub-sampled tasks at all but the highest values of $|\mathcal{H}|$. Refer to Appendix D.4 for additional details.

Table 2: Classification accuracy results in percent on VTAB+MD using Simple CNAPs and ProtoNets with varying values of $|\mathcal{H}|$. Image sizes are $224 \times 224$ and $84 \times 84$ pixels. For $|\mathcal{H}| > 40$, we used gradient/activation checkpointing methods [12] in addition to LITE. For full results see Appendix D.4

| Model | Simple CNAPs | | | | | | ProtoNets | | | | | Simple CNAPs | |
| Image Size | $224 \times 224$ | | | | | | $224 \times 224$ | | | | | $84 \times 84$ | |
| $|\mathcal{H}|$ | 1 | 10 | 20 | 30 | 40 | 100 | 0 | 10 | 20 | 30 | 40 | 40 | $|\mathcal{D}_S|$ |
|---|---|---|---|---|---|---|---|---|---|---|---|---|---|
| MD-v2 | 72.8 | 73.7 | 73.3 | 73.8 | 73.9 | 74.3 | 71.0 | 72.0 | 72.0 | 72.5 | 72.7 | 63.6 | 68.4 |
| VTAB (all) | 51.2 | 51.0 | 50.5 | 51.1 | 51.4 | 51.2 | 45.1 | 45.8 | 46.2 | 46.2 | 46.1 | 42.7 | 44.7 |
| VTAB (natural) | 64.5 | 65.3 | 64.1 | 65.8 | 65.2 | 66.0 | 58.5 | 60.0 | 60.6 | 60.8 | 60.9 | 47.7 | 49.5 |
| VTAB (specialized) | 71.8 | 71.4 | 70.5 | 71.3 | 71.9 | 71.6 | 63.5 | 63.9 | 64.2 | 64.5 | 64.2 | 61.0 | 63.8 |
| VTAB (structured) | 31.0 | 30.0 | 30.3 | 29.9 | 30.8 | 29.9 | 26.0 | 26.2 | 26.4 | 26.1 | 25.9 | 29.9 | 31.7 |

# 6  Discussion

We propose LITE, a general and memory-efficient episodic training scheme for meta-learners that enables them to exploit large images for higher performance with limited compute resources. LITE's significant memory savings come from performing a forward pass on a task's full support set, but back-propagating only a random subset, which we show is an unbiased estimate of the full gradient. We demonstrate that meta-learners trained with LITE are state-of-the-art among meta-learners on two challenging benchmarks, ORBIT and VTAB+MD, and are competitive with transfer learning approaches at a fraction of the test time computational cost.

This offers a counterpoint to the recent narrative that transfer learning approaches are all you need for few-shot classification. Both classes of approach are worthy pursuits (and will need to exploit large images in real-world deployments) but careful consideration should be given to the data and compute available at test time to determine which class is best suited to the application under consideration. If it involves learning just a single task type (e.g. classifying natural images) with ample data and no compute or time constraints, then a fine-tuning approach would suffice and perform well. However, if a multitude of task types will be encountered at test time, each with minimal data, and new tasks need to be learned on resource-constrained devices (e.g. a mobile phone or a robot) or quickly/repeatedly (e.g. in continual or online learning settings), then a meta-learning solution will be better suited.

Finally, as the machine learning community grapples with greener solutions for training deep neural networks, LITE offers a step in the right direction by allowing meta-learners to exploit large images without an accompanying increase in compute. Future work may look toward applying the basic concept of LITE to other types of training algorithms to realize similar memory savings.

**Limitations**  As discussed in Section 3, LITE can be applied to a wide range of meta-learners provided that they aggregate the contributions from a task's support set via a permutation-invariant operation like a sum. Because only a subset of the support set is back-propagated, however, the gradients can be more noisy and meta-training may require lower learning rates. Furthermore, LITE is a memory-efficient scheme for training meta-learners episodically and has not been tried with meta-learners trained in other ways (e.g. with standard supervised learning) or non-image datasets.

**Societal impact**  Few-shot learning systems hold much positive potential – from personalizing object recognizers for people who are blind [14] to rendering personalized avatars [30] (see [23] for a full review). These systems, however, also have the potential to be used in adverse ways – for example, in few-shot recognition in military/surveillance applications. Meta-trained few-shot systems may also pose risks in decision making applications as uncertainty calibration in meta-learning models has not yet been extensively explored. Careful consideration of the intended application, and further study of uncertainty quantification in meta-learning approaches will be essential in order to minimize any negative societal consequences of LITE if deployed in real-world applications.

# Acknowledgments

We thank the anonymous reviewers for key suggestions and insightful questions that significantly improved the quality of the paper. Additional thanks go to Vincent Dumoulin for providing the tabular results for SUR used in Fig. 3 and Table D.2.

## Funding Transparency Statement

Funding in direct support of this work: John Bronskill, Massimiliano Patacchiola and Richard E. Turner are supported by an EPSRC Prosperity Partnership EP/T005386/1 between the EPSRC, Microsoft Research and the University of Cambridge.

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
