# Appendix:
# Memory Efficient Meta-Learning with Large Images

## A  Applying LITE to meta-learners

### A.1  CNAPS, Simple CNAPS + LITE

We show the LITE processing flow for both meta-training and meta-testing phases of CNAPs/Simple CNAPs in Fig. A.1. For each query set meta-training batch $b$, the support set images $\{\boldsymbol{x}_{\mathcal{H}}, \boldsymbol{x}_{\overline{\mathcal{H}}}\}$ are broken into batches, passed through a 2D conv net, then coalesced so that the pooling step can compute the mean of all the support set embeddings. This mean embedding is then passed into the FiLM parameter generator so that the feature extractor can be configured for the task. The support set images $\{\boldsymbol{x}_{\mathcal{H}}, \boldsymbol{x}_{\overline{\mathcal{H}}}\}$ are then passed through the adapted feature extractor in batches and the outputs are coalesced and then fed along with the support set labels $\{y_{\mathcal{H}}, y_{\overline{\mathcal{H}}}\}$ into the box labeled "Compute Classifier Params". For CNAPs, this box performs the class-conditional pooling operation and then uses an MLP to generate the weights and biases for the linear classifier. For Simple CNAPs, the same box computes the class-conditional means and covariances that are then used by the classifier in the Mahalanobis distance calculations. Once the classifier has been configured, the images in the query set batch $\{\boldsymbol{x}_b^*\}$ can be classified and along with the true labels $\{y_b^*\}$, a loss is then computed. The meta-testing flow is similar, with the exception of the loss computation.

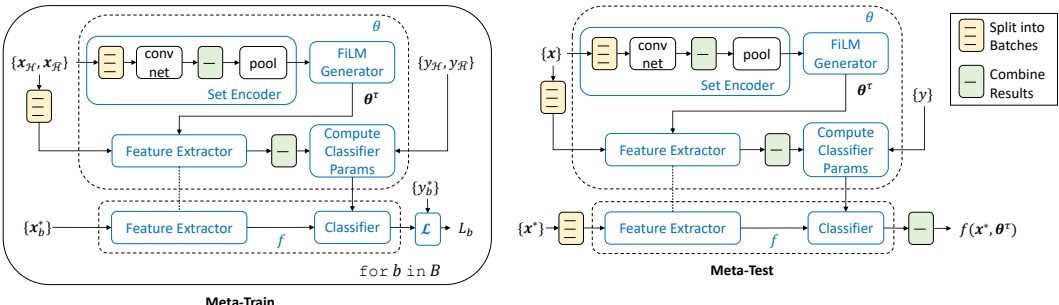

Figure A.1: CNAPs [4], Simple CNAPs [5] with LITE processing flow.

### A.2  ProtoNets + LITE

We show the LITE processing flow for both meta-training and meta-testing phases of ProtoNets in Fig. A.2. For each query set meta-training batch $b$, the support set images $\{\boldsymbol{x}_{\mathcal{H}}, \boldsymbol{x}_{\overline{\mathcal{H}}}\}$ are broken into batches, passed through the feature extractor and the resulting embeddings are then combined. The combined embeddings along with the support set labels $\{y_{\mathcal{H}}, y_{\overline{\mathcal{H}}}\}$ are then used to compute the class prototypes. The query batch images $\{\boldsymbol{x}_b^*\}$ are then passed through the feature extractor and the Euclidean distance from each query set image embedding to each of the class prototypes is computed. The predicted class is the one with the minimum distance. These predictions along with the true labels $\{y_b^*\}$ are used to compute the loss. The meta-testing flow is similar, with the exception of the loss computation.

## B  Additional Simple CNAPS details

Our implementation of Simple CNAPs differs slightly from [5]. Here we describe the key architecture differences which were made with the goal of reducing the number of model parameters. We verified that these modification came without a reduction in classification performance:

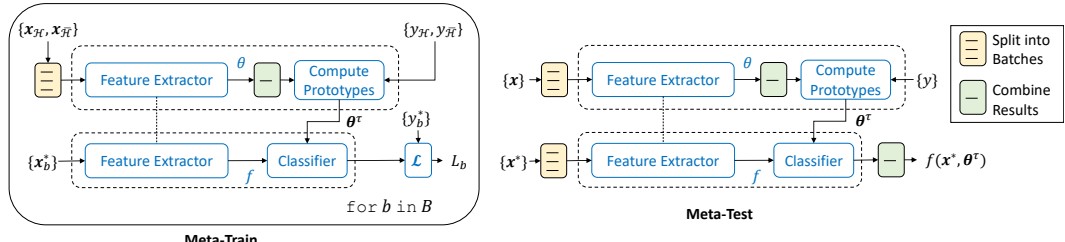

Figure A.2: ProtoNets [3] with LITE processing flow.

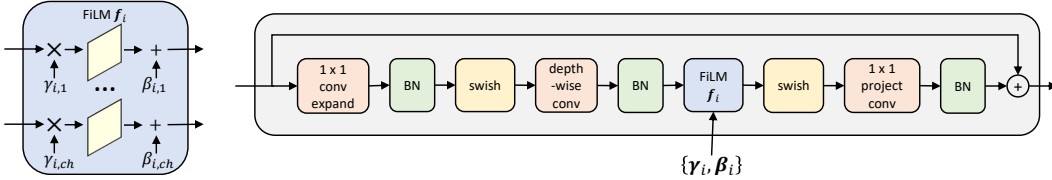

Figure B.3: (Left) A FiLM layer operating on convolutional feature maps indexed by channel $ch$. (Right) How a FiLM layer is used within a inverted residual block [33] of an EfficientNet [32].

- We replace the ResNet18 [31] feature extractor with an EfficientNet-B0 [32] since it has superior classification performance and fewer parameters (4.0M versus 11.2M for ResNet18). We pre-train the parameters of the feature extractor on ImageNet [29] and then freeze them during meta-training and meta-testing.

- Like Simple CNAPs, we use Feature-wise Linear Modulation (FiLM) layers [18] to adapt the feature extractor to the current task. In the EfficientNet-B0 feature extractor, we use a FiLM layer with scale parameters $\gamma_i$ and offset parameters $\beta_i$ after every separate convolutional layer and after every depth-wise separable convolution within a inverted residual block (refer to Fig. B.3). This is a total of 18 FiLM layers (<0.2% parameters in the model).

- We use a lower capacity 2-layer MLP network for generating parameters for each FiLM layer in the feature extractor (refer to Fig. B.4). This new FiLM layer generator network has less than 18% of the parameters (1.51M versus 8.45M) compared to the network used in the original Simple CNAPs.

- We do not use the Simple CNAPs Auto-regressive (AR) mode as the additional number of parameters did not yield sufficient gain.

Since the feature extractor parameters are frozen and the Mahalanobis distance based classifier has no parameters, the only learnable parameters in the model are in the set encoder and the network that generates the FiLM layer parameters.

## C   Experimental Details

In this section, we provide details for the LITE experiments using the ORBIT and VTAB+MD datasets.

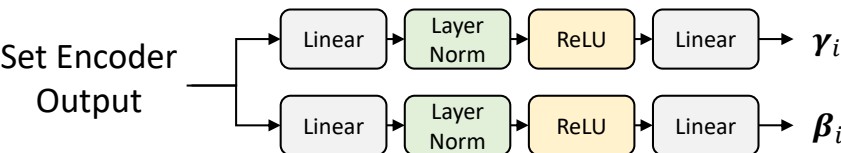

Figure B.4: Generator for $i^{th}$ FiLM layer. The generator takes the output of the set encoder step for each task $\tau$ and passes it through the network to generate the parameters $\gamma_i$ and $\beta_i$. The dimension of the vectors $\gamma_i$ and $\beta_i$ is equal to the number of feature channels at the location where the $i^{th}$ FiLM layer is placed within feature extractor. The depicted generator network structure is repeated for each FiLM layer added to the feature extractor.

## C.1 ORBIT Teachable Object Recognition Benchmark

Meta-training and meta-testing for the ORBIT experiments were performed on a single NVIDIA Titan RTX with 24GB of memory.

**Feature extractors**   We use either a ResNet-18 (following [14]) or an EfficientNet-B0 [32], both pre-trained on ImageNet [34]. Note, for CNAPs and Simple CNAPs, the feature extractor is frozen and only the set encoder and hyper-networks are trained, for ProtoNets and MAML all parameters are learned, and for the FineTuner the feature extractor is frozen and only the linear classifier is fine-tuned.

**Meta-training protocol**   We train the learnable parameters in the meta-learners episodically on 50 randomly sampled tasks per train user per epoch (44 total train users). Note, each epoch samples 50 *new* tasks per train user. Each task is composed of clips sampled from a single user's objects (random way) and associated videos (random shot). In the case of a large task, following [14], we randomly sample 4 clips from each support and query video, where each clip is 8 frames. For a small task, we limit this to 1 clip per support video and 1 clip per query video where each clip is 8 frames, and we also cap 1) the number of objects per task to 5, and 2) the number of support/query videos per object to 2. For both large and small tasks, a clip feature is taken as the average of its frame features, where each frame in $224 \times 224$ pixels. Note, the FineTuner undergoes no training – all feature extractor parameters are frozen to its pre-trained weights.

**Meta-testing protocol**   Following [14], we evaluate the trained models on 5 tasks per test user, where each task is sampled from just that user's objects and videos. Different to training, here each task contains *all* the test user's objects and associated videos without caps. For each test task, we randomly sample 8 clips from each support video, and *all* overlapping clips from each query video. We then adapt the trained model to a task by using the task's support clips to: i) perform a forward pass for CNAPs, Simple CNAPs and ProtoNets, ii) take 15 gradient steps on all the model's parameters for MAML, or iii) take 50 gradient steps on just the linear classifier head for FineTuner. We evaluate the adapted model predictions for every clip in each query video in the test task (in the clean video evaluation mode, the query videos show just one object on a clear surface, while in the clutter video evaluation mode, the query videos show the object in a multi-object/cluttered scene). We report all metrics averaged over a flattened list of all the query videos from all tasks from all test users (17 test users, 85 tasks in total), along with its corresponding 95% confidence interval.

**Optimization hyper-parameters**   For CNAPs, Simple CNAPs, and ProtoNets, we use the Adam optimizer [35] and a learning rate of $10^{-4}$. For MAML, we use Adam and a learning rate of $10^{-5}$ for the outer loop, and Stochastic Gradient Descent (SGD) and a learning rate of $10^{-3}$ for the inner loop (rates reduced by 0.1 for the feature extractor in both loops). For the FineTuner, we use SGD and a learning rate of 0.1. We train Simple CNAPs with ResNet-18/EfficientNet-B0 for 10/15 epochs respectively, CNAPs for 15/15 epochs, ProtoNets for 20/20 epochs, and MAML for 20/20 epochs. These were chosen based on the number of learnable parameters in each model. Table 1 reports the test performance of the model with the best frame accuracy on a held-out validation set.

**LITE hyper-parameters**   We train CNAPs, Simple CNAPs and ProtoNets with $H = 8$ clips (see Algorithm 1). We set the query batch size to $M_b = 8$ clips across all meta-learners. Note, MAML does not use LITE since we implement only the first-order variant. We, therefore, process support (and query) sets using standard batch processing with a batch size of 32 clips.

## C.2 VTAB+MD Benchmark

Meta-training and meta-testing for the VTAB+MD experiments were performed on a single NVIDIA V100 16GB GPU. Meta-training takes about 20 hours.

**Meta-training protocol**   Simple CNAPs + LITE uses an EfficientNet-B0 pretrained on ImageNet for the feature extractor $f$ and all of its parameters are frozen and not updated during meta-training. As permitted in the VTAB+MD protocol, we meta-train Simple CNAPs + LITE in an episodic manner on the training splits of following datasets: ImageNet, Omniglot, Aircraft, CU Birds, DTD, QuickDraw, and Fungi. In addition, we meta-train on the test split of MNIST as it does not overlap

with any of the test datasets. We meta-train for 10,000 iterations with the Adam [35] optimizer using a fixed learning rate of 0.001, and a batch size of 40. We back-propagate after every task, but do an optimization step after every 16 tasks.

**Meta-testing protocol**   For meta-testing on MD-v2, we generate test episodes using the Meta-Dataset episode reader with the standard evaluation settings. We test all models with 600 episodes each on all test datasets. The classification accuracy is averaged over the episodes and a 95% confidence interval is computed. For each test dataset in VTAB-v2, we use the TensorFlow Datasets API [36] and randomly sample 1000 examples from the train split for the support set and use the entire test split for the query set and report a single accuracy.

# D   Additional Experimental Results

## D.1   Full results on ORBIT benchmark

In the main paper, we report frame and video accuracy for test tasks, as well as the number of MACs and steps to adapt at test time and the number of model parameters. In Table D.1, we include a further metric – frames to recognition or FTR – which was proposed in the original baselines [14]. We also include additional results for large images (224) on small tasks without using LITE. Descriptions for the metrics are thus:

- **Frame accuracy**, the proportion of correct frame predictions in a query video;
- **Frames-to-recognition or FTR**, the number of frames before the first correct prediction, divided by the number of frames in the query video;
- **Video accuracy**, 1 if the most frequent frame prediction in a query video equals the true video label, otherwise 0;
- **MACs to adapt**, number of Multiply-Accumulate operations to learn a new task at test time (i.e. the operations required to process the whole support set);
- **Steps to adapt**, number of steps to learn a new task at test time (note, for gradient-based methods this involves multiple forward-backward passes through the model, while for amortization- and metric-based approaches this involves just a single forward pass);
- **Number of parameters**, number of learnable and frozen parameters in the model (note, this exclude the parameters that are generated by amortization-based methods)

## D.2   Tabular results on VTAB+MD benchmark

In Table D.2, we show the tabular results for the VTAB+MD benchmark.

## D.3   Meta-training without LITE on small tasks with large images

Table D.3 we show classification results on VTAB+MD using various ablations of Simple CNAPs including LITE on versus off, image size 84×84 versus 256×256 pixels, and small versus large sized tasks. For the no LITE, image size 84×84, and large task case, we meta-train on 35,000 tasks using the Adam optimizer at learning rate of 0.001 on the same training datasets as Simple CNAPs + LITE. For the no LITE, image size 224×224 pixels, and small task case, we we meta-train on 15,000 tasks using the Adam optimizer at learning rate of 0.001 on the same training datasets as Simple CNAPs + LITE. To make the number of tasks small during meta-training, we limit the maximum support set size to be 40 and the maximum classification way to be 30.

It is clear that using larger images results in a significant boost in classification accuracy, except on datasets where the images are natively small (e.g. Omniglot, Quickdraw, dSprites). Using LITE versus a smaller task size results in a significant boost in classification accuracy on VTAB-v2 where the support set size is large (1000 examples), however the results on MD-v2, where the support set sizes are smaller, are very similar in the two cases.

The trend is similar in the case of the ORBIT dataset (refer to Table D.1) where the difference between using LITE and tasks with a smaller number of examples is not great (often within the margin of error). This is likely due to the fact that in the case of ORBIT (i) the classification way is typically

Table D.1: Training meta-learners on large images with LITE achieves state-of-the-art accuracy with low test time adaption cost on the ORBIT Teachable Object Recognition Benchmark. Results are reported as the average (95% confidence interval) over 85 test tasks (5 tasks per test user, 17 test users). $I$ is image size. $f$ is model trained with/without LITE. RN-18 is ResNet-18. EN-B0 is EfficientNet-B0. T is $\times 10^{12}$ MACs. F is forward pass. FB is forward-backward pass. Time is average wall clock time per task in seconds.

| MODEL | $I$ $f$ | Clean Videos | | | Clutter Videos | | | Test-time adaption | | | |
|---|---|---|---|---|---|---|---|---|---|---|---|
| | | FRAME ACC ↑ | FTR ↓ | VIDEO ACC ↑ | FRAME ACC ↑ | FTR ↓ | VIDEO ACC ↑ | MACS ↓ | STEPS ↓ | TIME ↓ | PARAMS ↓ |
| FineTuner [28] | 84 RN-18 | 69.5 (2.2) | 7.8 (1.5) | 79.7 (2.6) | 53.7 (1.8) | 14.4 (1.5) | 63.1 (2.4) | 317.70T | 50FB | 53.94s | 11.17M |
| | 224 RN-18 | 72.2 (2.2) | 8.7 (17) | 81.9 (2.6) | 56.7 (2.0) | 18.8 (1.8) | 61.3 (2.5) | 546.57T | 50FB | 96.23s | 11.18M |
| | 224 EN-B0 | **78.1 (2.0)** | **5.8 (1.4)** | **85.9 (2.3)** | **63.1 (1.8)** | **11.5 (1.4)** | **66.9 (2.4)** | 121.02T | 50FB | 139.99s | **4.01M** |
| MAML [1] | 84 RN-18 | 70.6 (2.1) | 8.6 (1.6) | 80.9 (2.6) | 51.7 (1.9) | 21.0 (1.8) | 57.9 (2.5) | 95.31T | 15FB | 36.98s | 11.17M |
| | 224 RN-18 | 75.7 (1.9) | **4.9 (1.2)** | 86.1 (2.3) | 59.3 (1.9) | 16.3 (1.7) | 64.3 (2.4) | 163.97T | 15FB | 65.22s | 11.18M |
| | 224 EN-B0 | **79.3 (1.9)** | 6.2 (1.4) | **87.5 (2.2)** | **64.6 (1.9)** | **12.8 (1.5)** | **69.4 (2.3)** | 36.31T | 15FB | 117.89s | **4.01M** |
| ProtoNets [3] | 84 RN-18 | 65.2 (2.0) | 7.6 (1.4) | 81.9 (2.5) | 50.3 (1.7) | 14.9 (1.5) | 59.9 (2.5) | 3.18T | **1F** | **0.73s** | 11.17M |
| | 224 RN-18 | 77.4 (1.8) | **4.5 (1.1)** | 87.1 (2.2) | 56.8 (1.8) | 14.4 (1.5) | 62.5 (2.5) | 5.47T | 1F | 1.07s | 11.18M |
| | 224 RN-18 + LITE | 76.7 (1.9) | 5.1 (1.2) | 86.4 (2.2) | 61.4 (1.8) | **13.2 (1.5)** | **68.5 (2.4)** | 5.47T | 1F | 1.07s | 11.18M |
| | 224 EN-B0 | 78.4 (1.8) | 4.7 (1.1) | 87.9 (2.1) | 57.3 (1.8) | 12.7 (1.4) | 63.9 (2.4) | **1.21T** | 1F | 1.72s | **4.01M** |
| | 224 EN-B0 + LITE | **82.1 (1.7)** | **3.9 (1.0)** | **91.2 (1.9)** | **66.3 (1.8)** | 12.7 (1.5) | **72.9 (2.3)** | **1.21T** | 1F | 1.72s | **4.01M** |
| CNAPs [4] | 84 RN-18 | 66.2 (2.1) | 8.4 (1.4) | 79.6 (2.6) | 51.5 (1.8) | 17.9 (1.7) | 59.5 (2.5) | 3.48T | **1F** | 0.98s | 12.75M |
| | 224 RN-18 | 73.6 (2.0) | **5.4 (1.2)** | 83.4 (2.4) | 57.6 (1.8) | 14.9 (1.6) | 66.5 (2.4) | 7.64T | 1F | 2.11s | 12.76M |
| | 224 RN-18 + LITE | 76.0 (1.9) | 5.9 (1.3) | 84.9 (2.3) | 58.2 (1.9) | 15.1 (1.6) | 62.5 (2.5) | 7.64T | 1F | 2.11s | 12.76M |
| | 224 EN-B0 | **79.6 (1.9)** | 6.2 (1.4) | 87.0 (2.4) | 62.6 (1.9) | 13.2 (1.5) | 67.4 (2.4) | 3.38T | 1F | 2.83s | 10.59M |
| | 224 EN-B0 + LITE | **79.6 (1.9)** | 5.9 (1.3) | **87.6 (2.2)** | **63.3 (1.9)** | **12.8 (1.5)** | **69.2 (2.3)** | 3.38T | 1F | 2.85s | 10.59M |
| Simple CNAPs [5] | 84 RN-18 | 70.3 (2.1) | 7.3 (1.5) | 83.0 (2.5) | 53.9 (1.8) | 16.0 (1.6) | 62.0 (2.5) | 3.48T | **1F** | 1.01s | 11.97M |
| | 224 RN-18 | 75.2 (2.0) | 6.0 (1.4) | 84.6 (2.4) | 58.1 (1.9) | 14.7 (1.6) | 60.9 (2.5) | 7.64T | 1F | 2.13s | 11.97M |
| | 224 RN-18 + LITE | 76.5 (2.0) | 6.1 (1.4) | 86.4 (2.2) | 57.5 (1.9) | 17.3 (1.7) | 64.6 (2.4) | 7.64T | 1F | 2.14s | 11.97M |
| | 224 EN-B0 | 81.4 (1.8) | 4.9 (1.3) | 88.3 (2.1) | **65.6 (1.9)** | **11.2 (1.4)** | 69.9 (2.3) | 3.39T | 1F | 2.91s | 5.67M |
| | 224 EN-B0 + LITE | **82.7 (1.7)** | **4.1 (1.1)** | **91.8 (1.8)** | **65.6 (1.9)** | 13.5 (1.5) | **71.9 (2.3)** | 3.39T | 1F | 2.92s | 5.67M |

small (less than or equal to 10); and (ii) since the support frames are derived from videos, there is significant redundancy in the support sets, making the difference between having a small and large number of support examples less important. The benefits of LITE are more apparent in tasks with large way and large support set sizes, as is the case with VTAB-v2.

### D.4 Tabular results and additional details on the varying $|\mathcal{H}|$ experiments

Tables D.4 to D.6 provide full tabular results for classification accuracy versus varying $|\mathcal{H}|$ on VTAB+MD. Note that in Table D.6, using Simple CNAPs + LITE on images of size of $84 \times 84$ pixels with $|\mathcal{H}| = 40$, GPU memory usage drops to roughly 8 GB, which is approximately half of that used when run without LITE (i.e. $|\mathcal{H}| = |\mathcal{D}_S|$).

Table D.7 shows the mean squared error between the mean of the approximate gradients and the true gradients for both LITE and sub-sampled small tasks as $|\mathcal{H}|$ is varied. The low mean squared error values for both training methods empirically demonstrates that both are unbiased.

Table D.8 shows the average root mean squared error (RSME) of the approximate gradients and the true gradients for both LITE and sub-sampled small tasks as $|\mathcal{H}|$ is varied. Table D.8 and Fig. 4 show that the RMSE deviation of the LITE estimate is significantly smaller than that of sub-sampled small tasks at all but the highest values of $|\mathcal{H}|$. Note, that these results are limited to image classification in the specific networks and network parameters that we tested. Other data types and networks are left for future work.

These experiments were carried out as follows:

- The Simple CNAPs + LITE network is initialized identically for all runs.
- Image size is $84 \times 84$ pixels, so that the true gradients can be calculated.
- The same 10-way, 10-shot task ($|\mathcal{D}_S| = 100$) drawn from the DTD dataset is identical for all runs.
- Gradients are measured on the weights in the first (i.e. earliest) Conv2D layer in the set encoder after a single training iteration.
- Reference (exact) gradients are calculated without using LITE.
- Small task gradients are calculated by randomly sub-sampling the task (though we ensure there is at least one example per class).

Table D.2: Classification accuracy results on VTAB+MD [11] using Simple CNAPs + LITE and various competing transfer learning and meta-learning approaches. All competitive results are from [11]. All figures are percentages and the $\pm$ sign indicates the 95% confidence interval over tasks. Bold type indicates the highest scores (within the confidence interval). The VTAB-v2 results have no confidence interval as the testing protocol requires only a single run over the entire test set. RN indicates ResNet [31] and EN indicates EfficientNet [32]. The SUR results are with a linear classifier head. Simple CNAPs + LITE outperforms all approaches on MD-v2 and outperforms all meta-learning approaches on VTAB (all).

| | Transfer learning | | | Meta-Learning | | | | |
| | MD-Transfer | SUR | BiT | ProtoNets | ProtoMAML | CTX | SC(84) | SC+LITE |
|---|---|---|---|---|---|---|---|---|
| Backbone | RN-18 | RN-50 x 7 | RN-18 | RN-18 | RN-18 | RN-34 | EN-B0 | EN-B0 |
| Params (M) | 11.2M | 164.6M | 11.2M | 11.2M | 11.2M | 21.3M | 4.0M | 4.0M |
| Image Size | 126 | 224 | 224 | 126 | 126 | 224 | 84 | 224 |
| Omniglot | $82.0 \pm 1.3$ | 89.6 | $72.7 \pm 4.6$ | $85.3 \pm 0.9$ | $\mathbf{90.2 \pm 0.7}$ | $84.6 \pm 0.9$ | $\mathbf{90.9 \pm 0.6}$ | $86.5 \pm 0.8$ |
| Aircraft | $76.8 \pm 1.2$ | 59.7 | $73.6 \pm 3.8$ | $74.3 \pm 0.8$ | $82.1 \pm 0.6$ | $\mathbf{85.3 \pm 0.8}$ | $77.5 \pm 0.7$ | $83.6 \pm 0.7$ |
| Birds | $61.2 \pm 1.3$ | 81.4 | $\mathbf{87.2 \pm 1.9}$ | $68.0 \pm 1.0$ | $73.4 \pm 0.9$ | $72.9 \pm 1.1$ | $76.4 \pm 0.8$ | $\mathbf{88.6 \pm 0.7}$ |
| DTD | $66.0 \pm 1.1$ | 83.9 | $82.6 \pm 2.7$ | $65.3 \pm 0.7$ | $66.3 \pm 0.8$ | $77.3 \pm 0.7$ | $74.3 \pm 0.7$ | $\mathbf{84.1 \pm 0.7}$ |
| QuickDraw | $61.3 \pm 1.1$ | $\mathbf{81.2}$ | $66.3 \pm 3.6$ | $60.6 \pm 1.0$ | $66.4 \pm 1.0$ | $73.3 \pm 0.8$ | $76.5 \pm 0.7$ | $75.7 \pm 0.8$ |
| Fungi | $35.5 \pm 1.1$ | $\mathbf{69.2}$ | $53.9 \pm 4.4$ | $39.8 \pm 1.1$ | $46.3 \pm 1.1$ | $48.0 \pm 1.2$ | $51.3 \pm 1.1$ | $56.9 \pm 1.2$ |
| Traffic Sign | $\mathbf{84.7 \pm 0.9}$ | 46.5 | $75.4 \pm 4.3$ | $49.8 \pm 1.1$ | $50.3 \pm 1.1$ | $80.1 \pm 1.0$ | $54.8 \pm 1.1$ | $65.8 \pm 1.1$ |
| MSCOCO | $39.6 \pm 1.0$ | $\mathbf{58.6}$ | $60.0 \pm 2.9$ | $39.7 \pm 1.0$ | $39.0 \pm 1.0$ | $51.4 \pm 1.1$ | $45.1 \pm 1.0$ | $50.0 \pm 1.0$ |
| Caltech101 | 70.6 | 86.5 | 84.6 | 72.0 | 73.1 | 84.2 | 79.6 | $\mathbf{87.7}$ |
| CIFAR100 | 31.3 | 34.2 | 47.1 | 27.7 | 29.7 | 37.5 | 37.1 | $\mathbf{48.8}$ |
| Flowers102 | 66.1 | 71.2 | 82.7 | 57.1 | 60.2 | 81.8 | 65.5 | $\mathbf{83.5}$ |
| Pets | 49.1 | 88.7 | 83.9 | 51.0 | 56.6 | 70.9 | 69.8 | $\mathbf{89.3}$ |
| Sun397 | 13.9 | 0.5 | 29.1 | 14.2 | 8.1 | 24.8 | 18.0 | $\mathbf{30.9}$ |
| SVHN | 83.2 | 24.2 | $\mathbf{83.4}$ | 41.9 | 46.8 | 67.2 | 26.7 | 51.0 |
| EuroSAT | 88.7 | 82.6 | $\mathbf{93.8}$ | 77.7 | 80.1 | 86.4 | 82.8 | 89.3 |
| Resics45 | 63.7 | 67.8 | 74.1 | 50.8 | 53.5 | 67.7 | 64.5 | $\mathbf{76.4}$ |
| Patch Camelyon | $\mathbf{81.5}$ | 77.1 | 80.7 | 73.8 | 75.9 | 79.8 | 78.4 | 81.4 |
| Retinopathy | 57.6 | 37.4 | $\mathbf{74.5}$ | 28.0 | 73.2 | 35.5 | 29.4 | 40.3 |
| CLEVR-count | 40.3 | 34.1 | $\mathbf{55.2}$ | 32.0 | 32.7 | 27.9 | 30.7 | 31.4 |
| CLEVR-dist | 52.9 | 29.8 | $\mathbf{58.7}$ | 39.4 | 35.4 | 29.6 | 32.5 | 32.8 |
| dSprites-loc | 85.9 | 16.9 | $\mathbf{98.6}$ | 38.1 | 42.0 | 23.2 | 43.9 | 12.3 |
| dSprites-ori | 46.4 | 18.7 | 46.5 | 16.3 | 23.0 | $\mathbf{46.9}$ | 21.1 | 31.1 |
| SmallNORB-azi | 36.5 | 8.3 | 20.1 | 12.3 | 13.4 | $\mathbf{37.0}$ | 13.5 | 14.5 |
| SmallNORB-elev | $\mathbf{31.2}$ | 18.4 | 21.8 | 17.4 | 18.8 | 21.6 | 19.6 | 21.0 |
| DMLab | 43.0 | 33.5 | $\mathbf{43.7}$ | 31.8 | 32.5 | 31.9 | 33.9 | 39.4 |
| KITTI-dist | 58.7 | 57.5 | $\mathbf{78.8}$ | 42.1 | 54.4 | 54.3 | 58.1 | 63.9 |
| MD-v2 | 63.4 | 71.3 | 71.5 | 60.3 | 64.2 | 71.6 | 68.4 | $\mathbf{73.9}$ |
| VTAB (all) | 55.6 | 43.7 | $\mathbf{64.3}$ | 40.2 | 45.0 | 50.5 | 44.7 | 51.4 |
| VTAB (natural) | 52.4 | 50.9 | $\mathbf{68.5}$ | 44.0 | 45.7 | 61.1 | 49.5 | 65.2 |
| VTAB (specialized) | 72.9 | 66.2 | $\mathbf{80.8}$ | 57.6 | 70.7 | 67.3 | 63.8 | 71.9 |
| VTAB (structured) | 49.4 | 27.2 | $\mathbf{53.0}$ | 28.7 | 31.5 | 34.1 | 31.7 | 30.8 |

- For each value of $|\mathcal{H}|$, the number of samples used in the calculations are chosen such that 1000 examples of the support set are used. For example, for $|\mathcal{H}| = 50$, 20 different one iteration training runs are done (20 runs $\times$ 50 random examples per run = 1000).

- To calculate the values in Table D.7, for each value of $|\mathcal{H}|$, the mean of the approximate gradient runs is computed and then the mean squared error is computed between this value and the exact gradient.

- To calculate the values in Table D.8, for each value of $|\mathcal{H}|$, the RMSE between the approximate gradients and the exact gradients is computed and then this value is averaged over the number of runs.

## D.5 Additional Results

Table D.9 contains the results for Simple CNAPs + LITE on VTAB+MD at image size $320 \times 320$ pixel with $|\mathcal{H}| = 10$, demonstrating that by employing LITE, even larger images can be be used in meta-learning algorithms. Overall, these results are similar to the $224 \times 224$ case as the feature extractor was pre-trained at $224 \times 224$ pixels. However, on the Birds, Fungi, and Retinopathy datasets, where the original images are very large ($> 320$ pixels), the results on this run were better than the 224 case.

Table D.3: Classification accuracy results on VTAB+MD [11] using various ablations of Simple CNAPs including LITE on versus off, image size 84×84 versus 256×256 pixels, and small versus large tasks. All figures are percentages and the ± sign indicates the 95% confidence interval over tasks. Bold type indicates the highest scores (within the confidence interval). The VTAB-v2 results have no confidence interval as the testing protocol requires only a single run over the entire test set. A pretrained EfficientNet-B0 [32] backbone was utilized in all runs. In general, using larger images leads to better results, and using LITE on large tasks greatly improves results on VTAB-v2.

| LITE | No | No | Yes |
|---|---|---|---|
| Task Size | Large | Small | Large |
| Image Size | 84 | 224 | 224 |
| Omniglot | **90.9 ± 0.6** | **91.6 ± 0.6** | 86.5 ± 0.8 |
| Aircraft | 77.5 ± 0.7 | 81.5 ± 0.7 | **83.6 ± 0.7** |
| Birds | 76.4 ± 0.8 | **88.8 ± 0.6** | 88.6 ± 0.7 |
| DTD | 74.3 ± 0.7 | **83.7 ± 0.6** | 84.1 ± 0.7 |
| QuickDraw | **76.5 ± 0.7** | **76.4 ± 0.7** | 75.7 ± 0.8 |
| Fungi | 51.3 ± 1.1 | **59.3 ± 1.1** | 56.9 ± 1.2 |
| Traffic Sign | 54.8 ± 1.1 | 60.7 ± 1.0 | **65.8 ± 1.1** |
| MSCOCO | 45.1 ± 1.0 | **52.5 ± 1.1** | 50.0 ± 1.0 |
| Caltech101 | 79.6 | 84.9 | **87.7** |
| CIFAR100 | 37.1 | **50.2** | 48.8 |
| Flowers102 | 65.5 | 78.9 | **83.5** |
| Pets | 69.8 | 87.7 | **89.3** |
| Sun397 | 18.0 | **32.0** | 30.9 |
| SVHN | 26.7 | 37.6 | **51.0** |
| EuroSAT | 82.8 | 86.0 | **89.3** |
| Resics45 | 64.5 | 69.8 | **76.4** |
| Patch Camelyon | 78.4 | 79.1 | **81.4** |
| Retinopathy | 29.4 | 40.2 | **40.3** |
| CLEVR-count | 30.7 | 28.7 | **31.4** |
| CLEVR-dist | 32.5 | 31.4 | **32.8** |
| dSprites-loc | **43.9** | 14.7 | 12.3 |
| dSprites-ori | 21.1 | **35.8** | 31.1 |
| SmallNORB-azi | 13.5 | 12.2 | **14.5** |
| SmallNORB-elev | 19.6 | 19.0 | **21.0** |
| DMLab | 33.9 | 36.7 | **39.4** |
| KITTI-dist | 58.1 | 57.0 | **63.9** |
| MD-v2 | 68.4 | **74.3** | 73.9 |
| VTAB (all) | 44.7 | 49.0 | **51.4** |
| VTAB (natural) | 49.5 | 61.9 | **65.2** |
| VTAB (specialized) | 63.8 | 68.8 | **71.9** |
| VTAB (structured) | **31.7** | 29.4 | 30.8 |

Table D.4: Classification accuracy results on VTAB+MD [11] using Simple CNAPs + LITE with varying values of $|\mathcal{H}|$. Image size is 224 x 224 pixels. To achieve $|\mathcal{H}| > 40$, we used gradient/activation checkpointing methods [12] in addition to LITE. All figures are percentages and the $\pm$ sign indicates the 95% confidence interval over tasks. The VTAB-v2 results have no confidence interval as the testing protocol requires only a single run over the entire test set.

| Dataset | $|\mathcal{H}| = 1$ | $|\mathcal{H}| = 10$ | $|\mathcal{H}| = 20$ | $|\mathcal{H}| = 30$ | $|\mathcal{H}| = 40$ | $|\mathcal{H}| = 100$ |
|---|---|---|---|---|---|---|
| Omniglot | $83.5 \pm 1.0$ | $85.2 \pm 0.9$ | $85.5 \pm 0.9$ | $85.9 \pm 0.9$ | $86.5 \pm 0.8$ | $86.2 \pm 0.8$ |
| Aircraft | $82.1 \pm 0.8$ | $82.9 \pm 0.8$ | $82.5 \pm 0.8$ | $83.5 \pm 0.7$ | $83.6 \pm 0.7$ | $83.4 \pm 0.8$ |
| Birds | $88.0 \pm 0.7$ | $89.4 \pm 0.5$ | $88.9 \pm 0.6$ | $88.5 \pm 0.7$ | $88.6 \pm 0.7$ | $88.8 \pm 0.7$ |
| DTD | $84.4 \pm 0.7$ | $84.3 \pm 0.7$ | $84.2 \pm 0.7$ | $85.1 \pm 0.6$ | $84.1 \pm 0.7$ | $85.1 \pm 0.7$ |
| QuickDraw | $75.3 \pm 0.8$ | $75.8 \pm 0.8$ | $75.7 \pm 0.8$ | $75.9 \pm 0.8$ | $75.7 \pm 0.8$ | $76.1 \pm 0.8$ |
| Fungi | $53.8 \pm 1.2$ | $55.3 \pm 1.2$ | $56.8 \pm 1.2$ | $56.5 \pm 1.2$ | $56.9 \pm 1.2$ | $57.2 \pm 1.2$ |
| Traffic Sign | $65.9 \pm 1.1$ | $66.6 \pm 1.1$ | $64.7 \pm 1.1$ | $64.7 \pm 1.1$ | $65.8 \pm 1.1$ | $65.9 \pm 1.1$ |
| MSCOCO | $49.5 \pm 1.1$ | $50.0 \pm 1.1$ | $48.2 \pm 1.2$ | $50.2 \pm 1.1$ | $50.0 \pm 1.0$ | $51.9 \pm 1.1$ |
| Caltech101 | 87.9 | 87.8 | 87.1 | 87.5 | 87.7 | 88.2 |
| CIFAR100 | 46.8 | 46.6 | 45.2 | 48.1 | 48.8 | 50.1 |
| Flowers102 | 82.8 | 83.8 | 82.9 | 83.7 | 83.5 | 83.0 |
| Pets | 89.2 | 89.2 | 89.3 | 89.5 | 89.3 | 89.7 |
| Sun397 | 28.8 | 31.5 | 30.3 | 32.4 | 30.9 | 32.3 |
| SVHN | 51.4 | 53.0 | 49.6 | 53.5 | 51.0 | 52.7 |
| EuroSAT | 88.5 | 88.4 | 88.4 | 88.3 | 89.3 | 88.6 |
| Resics45 | 75.3 | 75.1 | 74.4 | 75.9 | 76.4 | 76.1 |
| Patch Camelyon | 80.4 | 80.2 | 78.7 | 80.2 | 81.4 | 81.9 |
| Retinopathy | 42.8 | 42.0 | 40.4 | 40.7 | 40.3 | 39.8 |
| CLEVR-count | 31.0 | 29.0 | 30.4 | 29.7 | 31.4 | 30.9 |
| CLEVR-dist | 33.4 | 33.4 | 32.6 | 32.8 | 32.8 | 33.0 |
| dSprites-loc | 10.7 | 10.5 | 11.6 | 11.3 | 12.3 | 10.6 |
| dSprites-ori | 30.5 | 29.9 | 29.4 | 29.2 | 31.1 | 27.9 |
| SmallNORB-azi | 14.6 | 14 | 14.2 | 14.3 | 14.5 | 14.6 |
| SmallNORB-elev | 21.3 | 21.3 | 20.8 | 20.7 | 21 | 20.9 |
| DMLab | 40.4 | 40.6 | 40.3 | 38.5 | 39.4 | 38.8 |
| KITTI-dist | 65.7 | 61.5 | 63.3 | 63.0 | 63.9 | 62.4 |
| MD-v2 | 72.8 | 73.7 | 73.3 | 73.8 | 73.9 | 74.3 |
| VTAB (all) | 51.2 | 51.0 | 50.5 | 51.1 | 51.4 | 51.2 |
| VTAB (natural) | 64.5 | 65.3 | 64.1 | 65.8 | 65.2 | 66.0 |
| VTAB (specialized) | 71.8 | 71.4 | 70.5 | 71.3 | 71.9 | 71.6 |
| VTAB (structured) | 31.0 | 30.0 | 30.3 | 29.9 | 30.8 | 29.9 |

Table D.5: Classification accuracy results on VTAB+MD [11] using ProtoNets with varying values of $|\mathcal{H}|$. Image size is 224 x 224 pixels. All figures are percentages and the $\pm$ sign indicates the 95% confidence interval over tasks. The VTAB-v2 results have no confidence interval as the testing protocol requires only a single run over the entire test set.

| Dataset | $|\mathcal{H}| = 0$ | $|\mathcal{H}| = 10$ | $|\mathcal{H}| = 20$ | $|\mathcal{H}| = 30$ | $|\mathcal{H}| = 40$ |
|---|---|---|---|---|---|
| Omniglot | $86.7 \pm 0.8$ | $87.7 \pm 0.8$ | $88.3 \pm 0.8$ | $88.3 \pm 0.7$ | $88.3 \pm 0.8$ |
| Aircraft | $83.8 \pm 0.7$ | $84.6 \pm 0.7$ | $84.1 \pm 0.7$ | $85.1 \pm 0.7$ | $85.0 \pm 0.7$ |
| Birds | $88.8 \pm 0.6$ | $89.4 \pm 0.6$ | $89.8 \pm 0.6$ | $89.1 \pm 0.7$ | $90.2 \pm 0.5$ |
| DTD | $78.6 \pm 0.6$ | $79.7 \pm 0.6$ | $80.2 \pm 0.7$ | $80.6 \pm 0.7$ | $81.4 \pm 0.6$ |
| QuickDraw | $73.5 \pm 0.8$ | $75.0 \pm 0.7$ | $75.2 \pm 0.8$ | $75.6 \pm 0.7$ | $76.0 \pm 0.7$ |
| Fungi | $59.4 \pm 1.2$ | $58.9 \pm 1.1$ | $58.2 \pm 1.2$ | $58.0 \pm 1.1$ | $57.4 \pm 1.1$ |
| Traffic Sign | $50.0 \pm 1.1$ | $52.2 \pm 1.1$ | $52.1 \pm 1.0$ | $53.1 \pm 1.1$ | $53.5 \pm 1.1$ |
| MSCOCO | $47.3 \pm 1.0$ | $48.1 \pm 1.0$ | $48.1 \pm 1.1$ | $50.2 \pm 1.0$ | $49.8 \pm 1.1$ |
| Caltech101 | 86.6 | 86.9 | 87.2 | 87.2 | 87.4 |
| CIFAR100 | 35.5 | 39.6 | 42.0 | 43.4 | 43.1 |
| Flowers102 | 76.6 | 77.9 | 78.3 | 78.5 | 78.2 |
| Pets | 88.5 | 88.4 | 88.7 | 88.7 | 88.6 |
| Sun397 | 31.5 | 31.8 | 31.1 | 31.8 | 32.9 |
| SVHN | 32.0 | 35.6 | 36.4 | 35.3 | 35.2 |
| EuroSAT | 79.4 | 81.6 | 81.8 | 82.9 | 83.3 |
| Resics45 | 65.7 | 67.4 | 68.0 | 69.0 | 68.8 |
| Patch Camelyon | 75.9 | 72.8 | 73.7 | 74.2 | 73.3 |
| Retinopathy | 32.9 | 33.7 | 33.2 | 31.9 | 31.3 |
| CLEVR-count | 28.3 | 27.3 | 27.1 | 27.2 | 27.2 |
| CLEVR-dist | 29.5 | 29.2 | 29.0 | 28.9 | 28.5 |
| dSprites-loc | 13.3 | 14.1 | 14.0 | 13.2 | 13.4 |
| dSprites-ori | 20.4 | 19.6 | 20.4 | 19.8 | 19.6 |
| SmallNORB-azi | 9.4 | 9.4 | 9.6 | 9.5 | 9.4 |
| SmallNORB-elev | 16.3 | 17.1 | 17.0 | 17.1 | 17.0 |
| DMLab | 35.2 | 35.5 | 35.9 | 35.9 | 35.8 |
| KITTI-dist | 55.6 | 57.2 | 58.2 | 57.1 | 56.5 |
| MD-v2 | 71.0 | 72.0 | 72.0 | 72.5 | 72.7 |
| VTAB (all) | 45.1 | 45.8 | 46.2 | 46.2 | 46.1 |
| VTAB (natural) | 58.5 | 60.0 | 60.6 | 60.8 | 60.9 |
| VTAB (specialized) | 63.5 | 63.9 | 64.2 | 64.5 | 64.2 |
| VTAB (structured) | 26.0 | 26.2 | 26.4 | 26.1 | 25.9 |

Table D.6: Classification accuracy results on VTAB+MD [11] using Simple CNAPS + LITE with two values of $|\mathcal{H}|$. Image size is 84 x 84 pixels. All figures are percentages and the $\pm$ sign indicates the 95% confidence interval over tasks. The VTAB-v2 results have no confidence interval as the testing protocol requires only a single run over the entire test set.

| Dataset | $|\mathcal{H}| = 40$ | $|\mathcal{H}| = |D_S|$ |
|---|---|---|
| Omniglot | 83.7±1.0 | 90.9±0.6 |
| Aircraft | 65.4±0.9 | 77.5±0.7 |
| Birds | 69.5±1.0 | 76.4±0.8 |
| DTD | 72.1±0.8 | 74.3±0.7 |
| QuickDraw | 70.6±0.9 | 76.5±0.7 |
| Fungi | 45.5±1.2 | 51.3±1.1 |
| Traffic Sign | 58.2±1.0 | 54.8±1.1 |
| MSCOCO | 43.8±1.1 | 45.1±1.0 |
| Caltech101 | 74.9 | 79.6 |
| CIFAR100 | 35.4 | 37.1 |
| Flowers102 | 69.6 | 65.5 |
| Pets | 55.5 | 69.8 |
| Sun397 | 13.9 | 18.0 |
| SVHN | 36.7 | 26.7 |
| EuroSAT | 84.2 | 82.8 |
| Resics45 | 61.5 | 64.5 |
| Patch Camelyon | 74.0 | 78.4 |
| Retinopathy | 24.3 | 29.4 |
| CLEVR-count | 32.2 | 30.7 |
| CLEVR-dist | 36.3 | 32.5 |
| dSprites-loc | 26.5 | 43.9 |
| dSprites-ori | 19.7 | 21.1 |
| SmallNORB-azi | 14.0 | 13.5 |
| SmallNORB-elev | 19.0 | 19.6 |
| DMLab | 33.4 | 33.9 |
| KITTI-dist | 58.1 | 58.1 |
| MD-v2 | 63.6 | 68.4 |
| VTAB (all) | 42.7 | 44.7 |
| VTAB (natural) | 47.7 | 49.5 |
| VTAB (specialized) | 61.0 | 63.8 |
| VTAB (structured) | 29.9 | 31.7 |

Table D.7: Mean Squared Error (lower is better) between the mean of the gradient estimates and the true gradients for both LITE and subsampled small tasks as $|\mathcal{H}|$ is varied. The task used was a 10-way, 10-shot task of $84 \times 84$ pixels from the DTD dataset. For each value of $|\mathcal{H}|$, 1000 support set examples were used.

| Training Mode | | | | | $|\mathcal{H}|$ | | | | |
|---|---|---|---|---|---|---|---|---|---|
| | 10 | 20 | 30 | 40 | 50 | 60 | 70 | 80 | 90 |
| LITE | 9.53E-11 | 9.24E-11 | 7.89E-11 | 8.48E-11 | 5.11E-11 | 5.31E-11 | 6.03E-11 | 1.02E-10 | 2.51E-11 |
| Subsampled Small Task | 9.23E-11 | 8.46E-11 | 7.67E-11 | 7.15E-11 | 6.45E-11 | 6.27E-11 | 5.67E-11 | 4.78E-11 | 4.30E-11 |

Table D.8: Average root mean squared error (lower is better) with respect to the exact gradients for both LITE and subsampled small tasks as $|\mathcal{H}|$ is varied. The task used was a 10-way, 10-shot task of $84 \times 84$ pixels from the DTD dataset. For each value of $|\mathcal{H}|$, 1000 support set examples were used.

| Training Mode | | | | | $|\mathcal{H}|$ | | | | |
|---|---|---|---|---|---|---|---|---|---|
| | 10 | 20 | 30 | 40 | 50 | 60 | 70 | 80 | 90 |
| LITE | 4.35E-03 | 3.20E-03 | 2.55E-03 | 2.32E-03 | 1.84E-03 | 1.63E-03 | 1.65E-03 | 1.73E-03 | 1.06E-03 |
| Subsampled Small Task | 5.56E-03 | 4.32E-03 | 3.43E-03 | 2.66E-03 | 2.37E-03 | 2.04E-03 | 1.77E-03 | 1.40E-03 | 1.13E-03 |

Table D.9: Classification accuracy results on VTAB+MD [11] using Simple CNAPs + LITE with $|\mathcal{H}| = 10$ and image size is $320 \times 320$ pixels. All figures are percentages and the $\pm$ sign indicates the 95% confidence interval over tasks. The VTAB-v2 results have no confidence interval as the testing protocol requires only a single run over the entire test set.

| Dataset | $\|\mathcal{H}\| = 10$, 320 x 320 pixels |
|---------|------------------------------------------|
| Omniglot | 83.2±1.0 |
| Aircraft | 82.5±0.8 |
| Birds | 91.2±0.6 |
| DTD | 85.3±0.7 |
| QuickDraw | 74.1±0.8 |
| Fungi | 58.0±1.2 |
| Traffic Sign | 62.4±1.1 |
| MSCOCO | 46.8±1.1 |
| Caltech101 | 88.0 |
| CIFAR100 | 45.2 |
| Flowers102 | 82.6 |
| Pets | 89.5 |
| Sun397 | 28.6 |
| SVHN | 51.9 |
| EuroSAT | 86.3 |
| Resics45 | 72.7 |
| Patch Camelyon | 80.7 |
| Retinopathy | 46.4 |
| CLEVR-count | 30.8 |
| CLEVR-dist | 33.5 |
| dSprites-loc | 14.2 |
| dSprites-ori | 28.2 |
| SmallNORB-azi | 14.0 |
| SmallNORB-elev | 20.7 |
| DMLab | 40.3 |
| KITTI-dist | 62.3 |
| MD-v2 | 72.9 |
| VTAB (all) | 50.9 |
| VTAB (natural) | 64.3 |
| VTAB (specialized) | 71.5 |
| VTAB (structured) | 30.5 |