# OpenReview forum: "Memory Efficient Meta-Learning with Large Images"
_NeurIPS.cc/2021/Conference — NeurIPS 2021 Poster_

### Official Review · Reviewer_ES9N · 2021-07-12

**Rating:** 6
**Confidence:** 3

**Summary:**

Many meta-learning approaches have been trouble with the scalability issue due to the large memory-required computation during the training process. To overcome such a problem, this paper proposes LITE that is memory efficient episodic training scheme allowing meta-learning methods to use the benefit of large-size images of tasks on a single GPU. On two benchmark datasets (ORBIT and VTAB+MB), meta-learning methods combined with LITE outperform meta-learning methods without LITE and Simple CNAPs (meta-learning method) + LITE shows competitive performance compared with transfer learning methods.

**Limitations And Societal Impact:**

Please refer to Comments in Main Review.

**Main Review:**

I think this paper is nice since it tackles the memory issue that has been a problem for a long time in the meta-learning field and the proposed solution is generally applicable for many meta-learning methods. However, I wish some clarity problems and analysis are improved during the rebuttal period.


- Strength
    - This paper is well written and easy to follow.
    - This paper tackles an important problem in the meta-learning domain, which is limited scalability due to the memory-intensive computation of training of support sets of tasks.
    - By overcoming the above issue, this paper achieves the SOTA performance on two few-shot benchmark datasets.


- Comments (Weakness)

    (order of importance)

    - Could the authors report memory savings and whether the performance drops for the same image sizes (e.g., 84) when we use LITE?
    - I wonder how much we can enlarge the image size or task size by using LITE with a single GPU?
    - In Table 1, could the authors report GPU hours (or Wall Clock Time) at the test time of each method? I wonder how much we can save time when we use the meta-learning methods compared with transfer learning as the authors described in the introduction.
    - In Table 1, could the authors report the results with 224 EN-B0?
    - Could the authors add discussion on [1] that is a method for large-scale meta-learning?

    [1] Shin et al. Large-Scale Meta-Learning with Continual Trajectory Shifting, ICML 2021.

**Time Spent Reviewing:**

5 hours

---

> ### Author Response · Authors · 2021-08-09
> **Response to Reviewer ES9N**
>
> Thanks for taking time to review our paper and for listing insightful comments and questions. Our responses are below:
>
> **1. “… report memory savings and whether the performance drops for the same image sizes (e.g., 84) when we use LITE?”**
>
> In a quick test using LITE with 84 x 84 images and $|\mathcal{H}|=40$, GPU memory drops by half to 8 GB, and classification accuracy drops slightly. Some learning rate tuning may make the differences smaller.
>
> | Test dataset                | LITE, $\|\mathcal{H}\|=40$  | No LITE  |
> |---------------------|:-----------:|:--------:|
> | MD-v2               |    63.6     |   68.4   |
> | VTAB (all)          |    42.7     |   44.7   |
> | VTAB (natural)      |    47.7     |   49.5   |
> | VTAB (specialized)  |    61.0     |   63.8   |
> | VTAB (structured)   |    29.9     |   31.7   |
> | Memory              |    ~8 GB    |  ~16 GB  |
>
> **2. “… how much we can enlarge the image size or task size by using LITE with a single GPU?”**
>
> By using $|\mathcal{H}|=10$, we were able to use 320 x 320 pixel images. The summary of the results on VTAB+MD are:
>
> | Test dataset | Accuracy  |
> |---------------------|:------:|
> | MD-v2               | 72.9  |
> | VTAB (all)          | 50.9  |
> | VTAB (natural)      | 64.3  |
> | VTAB (specialized)  | 71.5  |
> | VTAB (structured)   | 30.5  |
>
>
> Overall, these results are similar to the 224 x 224 case as the feature extractor was pre-trained at 224 x 224 pixels. However, on the Birds, Fungi, and Retinopathy datasets where the original images are very large ( > 330 pixels), the results on this run were better than the 224 case.
>
> **3. “… report GPU hours (or Wall Clock Time) at the test time of each method?”**
>
> In the table below we report the wall clock time to adapt to a task at test time on the ORBIT dataset. We will add these results to Table 1 as they are indeed insightful and further drive home the point that meta-learning methods are extremely light-weight and fast to adapt. This complements the “MACs to adapt” metric which has the added advantage of being independent of the GPU model and whether the testing job is competing with other processes for shared computer resources.
>
> | Model (all EN-B0)  | Time to adapt (s)  |
> |--------------------|:------------------:|
> | FineTuner          |       200.5        |
> | ProtoNets          |        5.5         |
> | CNAPs              |        6.5         |
> | Simple CNAPs       |        6.5         |
>
>
>
> **4. “In Table 1, could the authors report the results with 224 EN-B0?”**
>
> The results for 224 EN-B0 can be found in Table D.1 of the supplementary material as there was not enough room for them in the main paper. We plan to move them into Table 1 for the final version of the paper.
>
> **5. Could the authors add discussion on [1] that is a method for large-scale meta-learning?**
>
> Thanks for bringing this paper to our attention. Indeed, we will add a discussion on this paper in the final version of the paper. It would be interesting to compare [1] with the state-of-the-art BiT on the VTAB benchmark, which is a more competitive transfer learning approach than was used in the referenced paper. The largest image sized used in [1] was also  only 84 x 84 pixels, so it is unclear whether their method would efficiently scale to larger image sizes.
>
> **References:**
>
> [1] Shin et al. Large-Scale Meta-Learning with Continual Trajectory Shifting, ICML 2021.

---

> > ### Comment · Reviewer_ES9N · 2021-08-15
> > **Thank you for the answers.**
> >
> > Thank you for the explanation. All concerns are resolved. Could the authors add answers for **comment 1) and 3)** in the main paper or supplementary file?

---

> > > ### Author Response · Authors · 2021-08-16
> > > **Adding 1) and 3) to final manuscript.**
> > >
> > > Thank you for reading our response. Glad it clarified your questions. We will commit to adding responses to 1) and 3) to our manuscript.

---

### Official Review · Reviewer_qib5 · 2021-07-16

**Rating:** 6
**Confidence:** 3

**Summary:**

The authors propose a method that enables few-shot learners to be trained on 1 GPU without memory issues. This is very important since most few-shot learners need to process the whole train set of a task before evaluating the loss on the task. They show mathematically that their method gives an unbiased estimation of the true gradient (if all samples in training were back-propagated to update the model). They evaluate their method on CNAPS and ProtoNets and show that it can achieve better results since it can be trained with larger images.

**Limitations And Societal Impact:**

The authors described the limitations and societal impacts of their work.

**Main Review:**

The paper is well-written and well-structured. The problem that they try to solve is very important and also very challenging. The idea seems to be generalizable to different meta-learning techniques. They show it for CNAPS and ProtoNets. Their method is based on the idea to update the parameters of the model with the whole train set of the task but backpropagate through only a subset of the task. I think the mathematical study of this really helps the paper. Since this is mathematically correct, it might open new directions for new meta-learning algorithms.

I have two concerns. First, optimization-based algorithms like MAML that require several inner steps are still cannot get much improvement from this method. I looked at the supplementary material and the part that discussed MAML, however, I could not understand why it is not possible to limit the back-propagation on only a few samples from the train set. Can you please explain more about this?

My second concern is that some other related work like (Meta-learning with implicit gradients) can be used to solve the same problem. It seems that related work did not thoroughly cover other approaches for solving the memory issue in meta-learning. Has there been any other work that tackles this issue? I am aware that it might not be possible to compare directly, however, it is good to discuss them.

======Post rebuttal======

I thank the authors for clarifying reviewers' questions. I strongly suggest the authors modify the paper based on reviewers' requests. Especially, point 5 in reviewer wfg9's comment and experiments that were done for reviewer ES9N's comments. After discussing with other reviewers and reading their comments, I decided to stick with my initial review score of 6.

**Time Spent Reviewing:**

6

---

> ### Author Response · Authors · 2021-08-09
> **Response to Reviewer qib5**
>
> Thanks for taking the time to review our paper and asking good clarifying questions. Our responses are below:
>
> **1. …MAML… Can you please explain more about this?**
>
> As mentioned in our response to reviewer ijTZ, LITE does apply to MAML in theory (see equation 3 in the paper), but we did not implement it in practice. That said, LITE can offer considerable memory savings for full MAML as fewer Hessians would need to be computed on the support set. However, given that the computation graph from the inner loop computations (which would contain the activations for $|\mathcal{H}|$ examples from the support set) needs to be retained to do the outer loop update, it is not clear that the savings from LITE would be enough to allow the use of images of size 224 x 224 on a single GPU.
>
> To avoid the memory issues associated with the original MAML algorithm, several variations have been proposed, the most common is first-order MAML [1], where the computation graph from the inner loop steps on the support set is not retained and the implementation can use standard mini-batch processing methods to keep memory usage low. In this scenario, the use of LITE is not beneficial.
>
> **2. “…related work did not thoroughly cover other approaches for solving the memory issue in meta-learning.”**
>
> Thank you for pointing this out. Indeed, we do not adequately cover the several variations of the MAML algorithm that drastically reduce memory requirements such that LITE may not be useful or required (e.g. First-order MAML and the implicit gradient approach that you call out). For the final version of the paper, we will expand the related work section to include these methods.
>
> We also came across another deep learning memory reduction method -- gradient/activation checkpointing methods [2] -- after submitting the paper for review. These methods trade memory usage for additional computation to train on larger tasks at the expense of training time (i.e. during training they store only a subset of intermediate activations in a network needed for backpropagation and recompute the rest with additional forward computations when needed). We implemented this, but found that the memory savings still fell well short of what is needed to meta-train on the large task sizes required for large-scale benchmarks (e.g. VTAB+MD). We will add this information to the final version of the paper.
>
> **References:**
>
> [1] Finn, Chelsea, Pieter Abbeel, and Sergey Levine. "Model-agnostic meta-learning for fast adaptation of deep networks." International Conference on Machine Learning. PMLR, 2017.
>
> [2] Tianqi Chen, Bing Xu, Chiyuan Zhang, and Carlos Guestrin. Training deep nets with sublinear memory cost.arXiv preprint arXiv:1604.06174, 2016.

---

### Official Review · Reviewer_wfg9 · 2021-07-16

**Rating:** 6
**Confidence:** 4

**Summary:**

Meta-learning algorithms are not memory friendly since they have to load all the support samples during the inner-loop. When the support data are high-resolution images, or need huge support samples (i.e. 1000), the meta-training is hard to work on a single GPU. As a result, this paper aims to propose an efficient meta-training algorithm which only rely on the gradients of subsets of the support data. And they show the can achieve competitive results against transfer learning approaches but at a fraction of the test-time computational cost on multiple benchmarks.

**Limitations And Societal Impact:**

1)  Are the performance of the meta-learners sensitive to the choice of H?  Can we have a table or a figure about that, like accuracy v.s. different choice of H. I am interested in how meta-learning performs when H is the full support size or when H=0 (Just detach the gradient of the support samples).

2) Which one is more important to achieve a better performance? A larger support size or the gradient we needed for the support samples? Can we have the results where we have only the maximum number of support samples with 224 * 224 resolution to fit the memory?

3) The experiments only show that the proposed algorithm works, but lacks experiments like how this unbiased estimator performs against the full gradient. In addition, we need to show the gradient of the support samples are important.

**Main Review:**

The motivation of the paper is clear and the paper is easy to follow. The proposed algorithm is extremely simple and can be applied to almost all the meta-learners. The results on benchmarks which need large support samples seems promising. It is interesting to make the meta-learners more efficient but I am not sure how significant the problem is. In addition, I think more experiments are needed to show the effectiveness of the proposed algorithm.

**Time Spent Reviewing:**

2

---

> ### Author Response · Authors · 2021-08-09
> **Response to reviewer wfg9**
>
> Many thanks for reviewing our paper and providing great comments! We address your comments and questions below:
>
> **1. “It is interesting to make the meta-learners more efficient but I am not sure how significant the problem is.”**
>
> Using larger images for image classification tasks can yield a ten-percentage point gain when using images of size 224 x 224 versus size 96 x 96 pixels. This is a significant win (see Figure 5 in [1]) and a much more significant than many algorithmic improvements.
>
> Up until now, the majority of few-shot learning benchmarks have been limited to small images (e.g. miniImageNet and tieredImageNet use 84 x 84 pixel images, the majority of methods on Meta-Dataset use 126 x 126 pixels or less). While useful, small images are inherently a toy set-up and may be limiting the research insights/progress we can make. In addition, they limit the transferability of research models to real-world solutions (e.g. smart phone cameras acquire images in the mega-pixel range). Our paper takes steps to close this gap which we hope will lead to a much wider use of meta-learning methods in both research and practice.
>
> **2. “Are the performance of the meta-learners sensitive to the choice of H?”**
>
> Reviewer ijTZ suggested investigating the same idea. Please see our response to Reviewer ijTZ (point 2.). Note that the largest  $|\mathcal{H}|$ that can be used with LITE on a 16GB GPU is 40. If we combine LITE with gradient/activation checkpointing, we can get $|\mathcal{H}|$ up to 100, but no larger, which is still far short of the full possible support set size of 500 used when meta-training on VTAB+MD. That said, the overall sensitivity to $|\mathcal{H}|$ is relatively low.
>
> **3. “Which one is more important to achieve a better performance? A larger support size or the gradient we needed for the support samples?”**
>
> The results in Table D.3 (refer to the two rightmost columns) demonstrate that it is better to use a larger support set size (rightmost column) and sacrifice support set gradients as opposed to using a small support set with all the gradient information from the support set (second column from the right).
>
> **4. “Can we have the results where we have only the maximum number of support samples with 224 * 224 resolution to fit the memory?”**
>
> In the paper, we use the maximum $|\mathcal{H}|$ and image size that would fit in 16GB of memory when training on VTAB+MD (and 24GB when training on ORBIT). This amounts to  $|\mathcal{H}|=40$ on Meta-Dataset (and  $|\mathcal{H}|=64$ on ORBIT) and an image size of 224 x 224 pixels. Using all the gradient information (i.e. $|\mathcal{H}|=500$) is not possible on a 16GB GPU, even when combining LITE with gradient/activation checkpointing. If we reduce the image size to be 84 x 84 pixels, we can use the full support set size and get full gradients, but performance suffers (see the 84 column in Table D.3).
>
> **5. “…but lacks experiments like how this unbiased estimator performs against the full gradient.”**
>
> This is a good point. It is difficult to compute the full gradient because of the obvious memory limitations. We can, however, answer this question by considering smaller context set sizes. For the final revision of the paper, we will commit to do the following:
>
> For a small fixed support set $D_S$, we will repeat many times:
>
> - Compute the true gradient with all examples.
>
> - Select a value of $|\mathcal{H}|$ between 1 and $|D_S|$.
> - Compute the LITE estimate of the gradient with the value of $|\mathcal{H}|$.
> - Measure the L2 norm between the estimated gradient and the true gradient.
>
> Then we will plot the mean and the variance of the L2 norm difference as a function of $|\mathcal{H}|$. The LITE estimates will be unbiased and the variance’s dependence on $|\mathcal{H}|$ will be investigated.
>
> **6. “In addition, we need to show the gradient of the support samples are important.”**
>
> The support set examples are indeed important. Without them, the set encoder which aids in generating the FiLM layer parameters could not be trained.
>
> References:
>
> [1] Mark Sandler, Andrew Howard, Menglong Zhu, Andrey Zhmoginov, and Liang-Chieh Chen. Mobilenetv2: Inverted residuals and linear bottlenecks. In Conference on Computer Vision and Pattern Recognition (CVPR), pages 4510–4520, 2018.

---

> > ### Comment · Reviewer_wfg9 · 2021-08-27
> > **Thank you for the answers.**
> >
> > Thank you for the responses. Most of my concerns are addressed. I agree that the memory efficiency is an important problem for meta-learning which enables you to use high-resolution images during training. My concern is that how challenging is this problem, have you tried detaching gradient for the support data for those methods which do not have a FiLM layer such as ProtoNet? It is super interesting to me that the size of the subset of the gradient does not matter a lot, based on the table you provided for the different size of H, even when we use only 1 gradient. As a result, it seems to me that the gradient of the support data is not that important, we can achieve almost the same performance with a coarse estimation.

---

> > > ### Author Response · Authors · 2021-08-30
> > > **ProtoNets and No Support Set Gradient**
> > >
> > > Thanks for your response to our rebuttal. We are happy to hear that most of your concerns are now addressed. Regarding your remaining concern, we ran the following experiment using ProtoNets (with no FiLM layers) on VTAB+MD with the following results:
> > >
> > > | Test dataset | LITE, $\|\mathcal{H}\|=0$ | LITE, $\|\mathcal{H}\|=40$ |
> > > | --------- |:----------:|:-----------------:|
> > > | MD-v2 | 71.0 | 72.7 |
> > > | VTAB (all) | 45.1 | 46.1 |
> > > | VTAB (natural) | 58.5 | 60.9 |
> > > | VTAB (specialized) | 63.5 | 64.2 |
> > > | VTAB (structured) | 26.0 | 25.9 |
> > >
> > > As can be seen from the table above, the results at $|\mathcal{H}|=0$ are indeed respectable, though they fall short of what can be achieved at $|\mathcal{H}|=40$. Thus, the support gradient information does improve the solution, albeit by only a percentage point or two. It should be noted that in ProtoNets the adaptation network that processes the support set shares all its parameters with the feature extractor, and so in this case the adaptation network changes through meta-learning even though the gradients from the support set are neglected. If no parameters are shared between the adaptation network and the feature extractor, as is the case for CNAPs, then the adaptation network will not change at all during learning if the gradients from the support set are neglected.
> > >
> > > For reference, it would be ideal to compute the classification accuracy for $|\mathcal{H}|=|D_S|$ (i.e. using the true, full support set gradient which is equivalent to not using LITE at all). Due to memory constraints, we cannot compute the true support gradients at an image size of 224 x 224 pixels. However, we can compute the full support set gradients at image size 84 x 84. This was done in our response to Reviewer ES9N using CNAPs on VTAB+MD. The results are repeated below:
> > >
> > > |Test dataset | LITE,$\|\mathcal{H}\|=40$ | No LITE|
> > > | --------- |:----------:|:-----------------:|
> > > | MD-v2 |	63.6 | 68.4 |
> > > | VTAB (all) | 42.7 | 44.7 |
> > > | VTAB (natural) | 47.7 | 49.5 |
> > > | VTAB (specialized) | 61.0 | 63.8 |
> > > | VTAB (structured) | 29.9 | 31.7 |
> > >
> > > Here we see that the difference in accuracy between the true gradient (i.e. without using LITE) and using LITE at $|\mathcal{H}|=40$ is significant. From the first table, we have a decrease in accuracy from $|\mathcal{H}|=40$ to $|\mathcal{H}|=0$ and from the second table we have a significant decrease in accuracy from using the true gradient to $|\mathcal{H}|=40$. We also know LITE's performance will smoothly interpolate between these two values as $|\mathcal{H}|$ is increased from 40 to the size of the largest support set (at which point it becomes exact). This validates the LITE approach, where it is able to trade off GPU memory usage for classification accuracy by varying $|\mathcal{H}|$.
> > >
> > > In the next iteration of the paper, we are committed to computing and plotting the full range of classification accuracy versus  $|\mathcal{H}|$.

---

> > > > ### Comment · Reviewer_wfg9 · 2021-09-01
> > > > **Thanks for the response.**
> > > >
> > > > Thanks again for the response. I do agree that the problem discussed in the paper is important and the proposed method works under the memory constraints. I strongly recommend to put the tables mentioned in the response into the final version of the paper. I would like to raise my score as the concern is solved.

---

> > > > > ### Author Response · Authors · 2021-09-01
> > > > > **We will add the tables in the final version of the paper**
> > > > >
> > > > > Thanks for your positive response. We are pleased to hear that your concerns have been addressed. Indeed, we will include the tables in the finsl version of the paper. Thanks again for raising the issues!

---

> ### Author Response · Authors · 2021-08-18
> **Follow-up comment to reviewer wfg9**
>
> We invite reviewer wfg9 to add any further comments if they have. We are happy to engage in further discussion or provide additional clarifications if needed.

---

### Official Review · Reviewer_ijTZ · 2021-07-29

**Rating:** 6
**Confidence:** 4

**Summary:**

This paper considers the problem of training image classification for meta-learning/few-shot learning models with large images. The authors point out that many categories of meta-learning algorithms have difficulty using large images because of memory constraints brought up by computation of gradients computed during meta-training. They decompose 3 major categories of meta-learning models (CNAP, MAML, and Proto-Nets) into a common gradient update that is troublesome w/r/t memory. Their idea to make training more feasible is to use all the support set items in the forward pass but approximate the backward pass using a small random subsample of the examples in the support set. They compare training their versions of memory-efficient, large input meta-learning models against recent work on ORBIT (a few-shot object recognition task) and VTAB+MD (a newer benchmark for few-shot classification that is based on Meta-Dataset and VTAB datasets). On both benchmarks, they show that training using larger images using their proposed method improves performance.

**Ethical Concerns:**

No ethical concerns.

**Limitations And Societal Impact:**

Yes.

**Main Review:**

Originality: the paper is considering an interesting aspect of the meta-learning problem (holding constant the method but using larger input images) that as far as I know hasn't been considered before.

Quality: the proposed method and its motivating analysis seem sound. The experimental comparison is also fair and highlights the benefit of using the large image input based on their proposed approximation. However, I believe the experiments could've been expanded. Why not consider mini-ImageNet or tiered-ImageNet benchmarks in which most work uses 84x84 inputs? It would be interesting to see the benefit of larger inputs on those commonly used benchmarks. Additionally, I think an ablation experiment showing how the quality and speed of training varies with the approximation parameter H would also have been useful to evaluate the method. Lastly, I also think first-order maml with a larger input would be a useful baseline to see the effect of using larger inputs in a different way than what the authors propose.

Clarity: the paper is very well-written. It motives the problem well and has a clear description of the proposed solution. The experiments also seem to be conducted fairly and are informative.

Significance: I think the results in the paper could be useful to researchers as they show a way to improve quality of meta-learning models simply by increasing the image resolution. One concern I have is that the authors include MAML in the class of methods for which the proposed approximation applies but in the end, do not implement or show results for the method since "...because of the memory required to retain the inner loop’s computation graph, training may still need to be scaled to smaller images or smaller tasks." I am wondering if its actually useful to include discussion of MAML then in the paper if the paper doesn't have a concrete proposal/implementation of the approximation of second-order MAML that allows it to be trained on larger images?

**Time Spent Reviewing:**

4

---

> ### Author Response · Authors · 2021-08-09
> **Response to Reviewer ijTZ**
>
> Thanks for taking the time to review our paper and providing insightful feedback! We address your comments and questions below:
>
> **1. “Why not consider mini-ImageNet or tiered-ImageNet benchmarks …”?**
>
> The primary reason we do not evaluate on these datasets is that the feature extractor in our meta-learners is pretrained on ILSVRC 2012 (i.e. ImageNet) and as a result it would be unfair to test benchmarks based on ImageNet, such as minImageNet or tiered-ImageNet. The secondary reason is because these datasets have been shown to provide a very limited test of a meta-learning method’s ability to generalize [1]. As a result, we chose to evaluate on more challenging, higher-scale benchmarks (i.e. VTAB+MD, ORBIT).
>
> **2. Please provide “…an ablation experiment showing how the quality and speed of training varies with the approximation parameter  $|\mathcal{H}|$ would also have been useful…”**
>
> Excellent idea. Thanks for the suggestion. We ran this and here are the results. We’ll add proper plots to the final version of the paper.
>
> ***Simple CNAPs results on VTAB + MD***
>
> | Benchmark / $\|\mathcal{H}\|$         | 1     | 10    | 20    | 30    | 40    | 100*  |
> |---------------------|-------|-------|-------|-------|-------|-------|
> | MD-v2               | 72.8  | 73.7  | 73.3  | 73.8  | 73.9  | 74.3  |
> | VTAB (all)          | 51.2  | 51.0  | 50.5  | 51.1  | 51.4  | 51.2  |
> | VTAB (natural)      | 64.5  | 65.3  | 64.1  | 65.8  | 65.2  | 66.0  |
> | VTAB (specialized)  | 71.8  | 71.4  | 70.5  | 71.3  | 71.9  | 71.6  |
> | VTAB (structured)   | 31.0  | 30.0  | 30.3  | 29.9  | 30.8  | 29.9  |
>
> *To achieve  $|\mathcal{H}| > 40$ for Simple CNAPs, we used gradient/activation checkpointing methods [2] in addition to LITE.
>
> ***ProtoNets results on ORBIT Clean***
>
> | Model / $\|\mathcal{H}\|$            |       8       |      16       |      32       |      64       |   |   |
> |--------------------|:-------------:|:-------------:|:-------------:|:-------------:|---|---|
> | ProtoNets (EN-B0)  | 82.34 (1.65)  | 82.78 (1.63)  | 83.12 (1.60)  | 83.12 (1.61)  |   |   |
>
>
> The results indicate that for both Simple CNAPs and ProtoNets the results are consistent and robust for different settings of  $|\mathcal{H}|$ which is expected as LITE provides an unbiased estimate of the true gradient and we meta-train on thousands of tasks.
>
> **3. “first-order MAML with a larger input would be a useful baseline”**
>
> Indeed, another great idea for other researchers to build off. We will include this baseline in the final version of the paper.
>
> **4. Is it “useful to include discussion of MAML then in the paper”?**
>
> This is a fair point. While LITE does apply to MAML in theory (see equation 3 in the paper), we don’t demonstrate its viability experimentally. In addition, it is not clear that a LITE version of MAML would have any advantages over existing memory efficient variants of MAML (e.g. First-order MAML [3] or Implicit Gradients [4]). We will reduce the MAML content in the final version of the paper and add the above context.
>
> **References:**
>
> [1] Huang, Gabriel, Hugo Larochelle, and Simon Lacoste-Julien. "Are Few-shot Learning Benchmarks Too Simple?.", arXiv preprint arXiv:2107.01105, (2019).
>
> [2] Tianqi Chen, Bing Xu, Chiyuan Zhang, and Carlos Guestrin. Training deep nets with sublinear memory cost.arXiv preprint arXiv:1604.06174, 2016.
>
> [3] Finn, Chelsea, Pieter Abbeel, and Sergey Levine. "Model-agnostic meta-learning for fast adaptation of deep networks." International Conference on Machine Learning. PMLR, 2017.
>
> [4] Rajeswaran, Aravind, et al. "Meta-learning with implicit gradients.", arXiv preprint arXiv:1909.04630 (2019).

---

### Author Response · Authors · 2021-08-09
**Thanks to all the reviewers!**

We thank reviewers for their positive feedback on our paper which contributes:
1. LITE, a mathematically justified, general, and memory-efficient episodic training scheme for meta-learners which enables meta-learners to be trained on large images on a single GPU, and
2. state-of-the-art accuracy for meta-learners on two of the most challenging few-shot learning benchmarks.

All the reviewers agreed that our work tackles a long-standing limitation in meta-learning research (i.e. reducing the memory required during training) with a simple and effective solution. They commended the generalizability of the approach [wfg9, qib5], the soundness of the mathematical justification and experiments [wfg9, qib5], and the paper's clarity/structure [ijTZ, wfg9, qib5, ES9N].

Primarily, the reviewers requested additional experiments and clarifications. Thus, in the rebuttal, we have performed 4 new experiments that demonstrate:

• The robustness of accuracy as the number of back-propagated examples (H) is varied [ijTZ, wfg9].

• The performance and memory usage of LITE on images sizes both smaller and larger than 224 x 224 pixels [ES9N].

• The wall clock timings on the ORBIT experiments [ES9N].

We believe that these experiments and our responses below address all questions, issues, and comments. We have also committed to performing two other clarifying experiments in the final version of the paper.

---

### Decision · Program_Chairs · 2021-09-27

**Decision:**

Accept (Poster)

**Comment:**

The paper is addressing the problem of meta-learning with large-scale images. In this setting, many of the existing algorithms require an intractable amount of memory. The proposed method addresses this challenge by simply sub-sampling the support set in the backward pass. Although the method is sensible, the empirical gain is marginal. All reviewers agree on the merit of the paper to be accepted after an active discussion phase.